# The Medical Genome Reference Bank contains whole genome and phenotype data of 2570 healthy elderly

Mark Pinese et al.[#]

Population health research is increasingly focused on the genetic determinants of healthy ageing, but there is no public resource of whole genome sequences and phenotype data from healthy elderly individuals. Here we describe the first release of the Medical Genome Reference Bank (MGRB), comprising whole genome sequence and phenotype of 2570 elderly Australians depleted for cancer, cardiovascular disease, and dementia. We analyse the MGRB for single-nucleotide, indel and structural variation in the nuclear and mitochondrial genomes. MGRB individuals have fewer disease-associated common and rare germline variants, relative to both cancer cases and the gnomAD and UK Biobank cohorts, consistent with risk depletion. Age-related somatic changes are correlated with grip strength in men, suggesting blood-derived whole genomes may also provide a biologic measure of age-related functional deterioration. The MGRB provides a broadly applicable reference cohort for clinical genetics and genomic association studies, and for understanding the genetics of healthy ageing.

---

[#]A full list of authors and their affiliations appears at the end of the paper.

Most developed nations face crises in health care associated with population ageing. Healthy ageing is a complex phenotype, influenced by both environmental and genetic factors. Healthy ageing—the absence of clinically significant, non-communicable disease or morbidity in old age —is distinct from longevity, which disregards quality of life. Healthy ageing captures the critical distinction between a long life with minimal impairment, and one bearing significant, costly, and potentially prolonged morbidity.

Relatively little is known about the genetic determinants of ageing that account for the broad spectrum of health states observed in older people. Genetic variation contributes to healthy ageing through pleiotropic effects on many diseases, immune responses, anthropomorphic, and behavioural phenotypes. For example, alleles associated with behavioural phenotypes that contribute to a healthy lifestyle, such as avoidance of smoking, or propensity for regular exercise, might be anticipated to have an effect on healthy ageing. However, to date, common variation at only a relatively small number of loci has been consistently associated with lifespan or parental longevity[1–3]. Rare pathogenic variants that hasten the onset of common diseases associated with age, such as cancer, cardiovascular disease, and neurodegenerative disorders, might also be expected to be depleted in healthy aged individuals. While the single study using whole-genome sequencing (WGS) in 511 healthy aged individuals confirmed the link with the *APOE* locus, and suggested depletion of poly-genic risk for Alzheimer's disease and coronary artery disease[4], no evidence was found for depletion of rare pathogenic variation. Limited by sample size, these studies have focussed on single-nucleotide variants and indels, while large-scale structural varia-tion remains unexplored. In addition, somatic variation such as clonal haematopoiesis is known to correlate with both age and susceptibility to disease[5,6]. A synthesis of all forms of somatic and germline genomic variation is needed to inform our under-standing of healthy ageing and disease susceptibility.

The advent of WGS is driving intense interest in mapping the genetic basis of disease, less than 50% of which is currently under-stood[7]. The missing heritability arguably resides in the total burden of both common and rare variation, structural variation untagged by simple polymorphisms, and their interactions[8,9]. WGS enables more comprehensive characterisation of common, rare, and complex variation in human cohorts. The next few years will see the release of large-scale WGS studies in rare diseases and cancer, such as the 100,000 Genomes Project, and population studies like the UK Bio-bank. Maximising the analytic power of whole-genome association studies using these cohorts will require well-phenotyped and high-quality control data. The concept of extreme phenotype sampling maximises statistical power by comparing the extremes of pheno-types of interest[10]. We postulate that an elderly cohort depleted of the major common diseases constitutes a powerful and broadly applicable tool for genome-wide association studies of disease.

With this background, we are undertaking WGS of over 4000 elderly individuals with no reported history of cancer, cardio-vascular disease, or neurodegenerative diseases up to age 70, to create the Medical Genome Reference Bank (MGRB)[11]. Here we describe the first release of this resource, comprising WGS data and phenotype for 2570 well elderly individuals. For comparison, we also perform WGS of 344 young subjects, and 273 elderly individuals with cancer. We subject these cohorts to a broad spectrum, systematic analysis of germline and somatic variation within the nuclear and mitochondrial genomes, which we link to both chronologic age as well as frailty measures.

## Results
### Cohort characteristics and sequencing
The MGRB consists of over 4000 individuals from the ASPREE study[12], and Sax

**Table 1 Summary metrics for the first release of the MGRB well elderly cohort.**

| Measure | ASPREE | 45 and Up |
|---|---|---|
| Individuals (percent female) | 1853 (48.2%) | 717 (59.3%) |
| Age at blood draw (years) | 79 (75–95) | 70 (64–91) |
| Height (m) | 1.65 (1.33–1.91) | 1.66 (1.37–1.91) |
| Mass (kg) | 74.5 (33.4–127.1) | 72.0 (36.0–147.0) |
| Mean sequencing depth (genome-wide) | 38.0 (26.8–46.0) | 39.0 (27.3–45.5) |
| Genetic background | | |
| Non-Finnish European | 1805 | 695 |
| South and Central American | 23 | 5 |
| South Asian | 14 | 6 |
| Finnish European | 10 | 7 |
| East Asian | 1 | 4 |

Samples were sourced from the ASPREE or 45 and Up studies. Aggregate statistics are medians, with ranges in parentheses. Genetic background (ancestry) was determined from genotype data Although blood was occasionally drawn at younger than 70 years, all individuals lived to at least 70 years without known cancer, cardiovascular disease, or dementia

Institute's 45 and Up Study[13], who lived to at least 70 years of age without any history of cancer, cardiovascular disease, or dementia, confirmed either at baseline entry or study follow-ups[11]. We sequenced blood of 2926 MGRB individuals by WGS, mapping to build 37 of the human reference genome, and calling variants following GATK best practices. After exclusion of 356 samples that failed quality control and relatedness checks, 2570 samples remained, forming the first release of the MGRB cohort genomic data (Table 1).

A broad diversity of genetic variation was found in the MGRB cohort. We identified 69,996,670 small variant loci in canonical chromosome contigs, with a call rate of 99.5%. Our small variant detection sensitivity was 99.3% and false-positive rate 4.84 Mbp, as assessed by comparing an internal RM 8398 sample against a gold standard[14]. MGRB participants were primarily of non-Finnish European genetic background (Table 1, Supplementary Fig. 1). Consistent with previous studies[15,16], 51.8% of small variants were singletons and 4.6% of loci were multi-allelic.

In addition to small scale variants, an average of 4036 structural variants (SVs) per individual were observed, most commonly deletions (Supplementary Table 1, Supplementary Fig. 2). In contrast to small variants, only 17% of SVs were unique (Supplementary Table 2). Each individual carried an average of 4264 mobile element insertions (MEI), predominantly of the ALU and L1 classes, and most MEIs were copy number polymorph-isms at known loci. However, on average 1535 MEI events per individual were in regions of the reference genome not currently described as containing mobile elements. In summary, while small variants comprise the majority of genetic diversity in the MGRB, structural and mobile elements constitute a rich and understudied source of potentially disease-related variation.

**Well elderly carry clinically reportable genetic variation.** Population genomic studies are contributing to the substantial revision of clinical interpretation of genetic variation thought to drive disease in some cases[17]. It is therefore clinically important to understand the frequency of variants currently considered pathogenic in a clinical context, but which are observed in well elderly individuals. To this end, we identified pathogenic var-iants that are considered clinically reportable as incidental findings under current American College of Medical Genetics

**Table 2 Counts of clinically significant small variation in the MGRB for all genes in the ACMG SF 2.0 set.**

| Condition | Gene | Carriers |
|---|---|---|
| Cancer | BRCA2 | 4 (2 female) |
| | MSH2 | 1 |
| | MSH6 | 1 |
| | PMS2 | 3 |
| Neurofibromatosis | NF2 | 1 |
| ARVC | DSG2 | 1 |
| | DSP | 3 |
| CPVT | RYR2 | 1 |
| HCM, DCM | MYBPC3 | 2 |
| | MYL3 | 1 |
| | TNNI3 | 1 |
| Hypercholesterolaemia | APOB | 5 |
| Long QT, VA | KCNH2 | 1 |
| | SCN5A | 1 |
| Marfan syndrome | MYH11 | 1 |
| Malignant hyperthermia | RYR1 | 1 |
| Total | | 28 |

*ARVC* arrhythmogenic right ventricular cardiomyopathy, *CPVT* catecholaminergic polymorphic ventricular tachycardia, *HCM* hypertrophic cardiomyopathy, *DCM* dilated cardiomyopathy, *VA* ventricular arrhythmia

(ACMG) guidelines[18]. Forty pathogenic or likely pathogenic heterozygous small variants were identified, with 28/2570 (1.1%) individuals carrying dominantly acting variants linked to disease (Table 2, Supplementary Data 1). We sought further evidence of disease phenotypes in individuals carrying relevant pathogenic variants from the ASPREE cohort. We did not identify personal histories of breast or colorectal cancer in individuals harbouring *BRCA2*, *MSH2*, or *PMS2* mutations; cardiac arrest or strokes in individuals harbouring *DSG2*, *DSP*, *KCNH2*, *KCNQ1*, *MYBPC3*, *MYL3*, and *SCN5A* mutations; or elevated blood lipid levels in *APOB* carriers. Cancer-associated genotypes are dependent on stochastic factors which may account for variable penetrance, while anaesthetic-associated malignant hyperthermia linked to loss-of-function variation in *RYR1* is contingent on environmental exposure. We specifically sought, but did not find, evidence of cardiovascular disease history or related clinical phenotypes in carriers of variants linked to atrial fibrillation, cardiomyopathy, and hypertension. Notably, no genotypes predicted to cause severe childhood-onset diseases were identified[19]; the single *RYR2* variant detected was a truncation not expected to cause autosomal dominant catecholaminergic polymorphic ventricular tachycardia. In five individuals, variants were noted in *PCSK9* that are predicted to be protective against high blood cholesterol[20]. Four SVs were found that may disrupt the coding sequence of genes associated with cancer and cardiovascular health (Supplementary Table 3), comprising 10% of potentially pathogenic variation in genes considered reportable by the ACMG.

**Risk variants are depleted in the well elderly.** One of the primary purposes of the MGRB is to serve as a genetic risk-depleted control cohort for studies of the common causes of morbidity and mortality. To test its utility, we compared the rates of pathogenic variants in tumour suppressor genes between the 717 MGRB individuals from the 45 and Up Study, with 269 demographically matched cancer cases from the same study (45 and Up Study; Supplementary Table 4). Considering all cancers in aggregate, the MGRB samples were significantly depleted for pathogenic alleles in tumour suppressor genes relative to cancer cases, with 2 of 717 controls carrying pathogenic tumour suppressor variants compared to 12 of 269

cancer cases (Fig. 1a, odds ratio 0.060, 95% confidence interval 0.0065–0.27, two-sided $p < 0.001$, $n = 986$, Fisher's exact test). In addition to all cancers, we specifically examined colorectal cancer due to its high incidence in our case set, and well-defined genetic risk. The MGRB samples were significantly depleted for rare pathogenic variation in the *APC*, *MLH1*, *MSH2*, *MSH6*, *PMS2*, and *SMAD4* genes, relative to colorectal cancer cases (Fig. 1a, 1 of 717 MGRB with pathogenic variants, vs 2 of 40 cancer cases, odds ratio 0.027, 95% CI 0.001–0.53, two-sided $p = 0.008$, $n = 757$, Fisher's exact test).

We next sought evidence for depletion of common disease-associated variation in the MGRB, relative to the gnomAD and UKBB datasets. Although SNP allele frequencies were highly concordant across all three cohorts (Supplementary Fig. 3), the MGRB cohort was significantly depleted for alleles specifically associated with risk of cancer, cardiovascular disease, and neurodegenerative disease (Supplementary Data 2, sheet 3, 698 loci, vs gnomAD odds ratio 0.38, 95% CI 0.27–0.52, $p = 2.6 \times 10^{-9}$; vs UKBB odds ratio 0.47, 95% CI 0.34–0.64, two-sided $p = 1.1 \times 10^{-6}$, Fisher's exact test). This enrichment of protective alleles was specific to the clinical phenotypes excluded from MGRB (cancer, cardiovascular disease, and dementia, see Methods), and was not observed for negative control loci linked to anthropometric (449 loci, both $p > 0.69$) or behavioural (575 loci, both $p > 0.55$) traits.

The aggregate burden of common disease-related variants within individuals can be summarised in a polygenic score (PS). We constructed polygenic predictors for a range of phenotypes measured or depleted in the MGRB, and compared PS between MGRB, the gnomAD non-Finnish European reference cohort, the UK Biobank (UKBB), and the 45 and Up Study cancer cohort. Since the MGRB is an Australian cohort in which allele frequencies differ slightly from the white British sample represented by the UKBB, we controlled for expected subtle differences in allele frequencies genome-wide by simulating the reference population as a derivative from the UKBB that differs due to drift by an amount typical of the MGRB-UKBB comparison. Similar comparisons were made for the gnomAD and 45 and Up comparison cohorts, and 100,000 bootstraps were performed (Supplementary Fig. 4). We observed significant depletion of PS in MGRB for 7 of the 12 scores tested (Fig. 1b, Supplementary Data 2, Sheet 9). Notably, a PS associated with short parental lifespan[1] was significantly depleted in MGRB relative to UKBB, consistent with the MGRB healthy elderly phenotype (bootstrap test, 0/100,000 rounds with a test statistic more extreme than observed, $p < 0.001$ after Holm correction for multiple tests). MGRB individuals were also depleted for prostate cancer risk relative to both UKBB (bootstrap test, 146/100,000 rounds more extreme than observed, Holm-corrected $p = 0.055$) and prostate cancer cases (two-sample $t$-test, $n = 71$ cases, $n = 278$ MGRB gender- and cohort-matched controls, $t = 3.96$, df $= 126$, two-sided $p < 0.001$), indicating that MGRB is an extreme depletion cohort for prostate cancer polygenic risk (Fig. 1c). Critically, for the extreme phenotype sampling hypothesis, the use of the MGRB as a control cohort reduced the sample size required to reach a given target power by approximately 25% by comparison with the UKBB (Fig. 1d).

In addition to the allele frequency-based comparisons above, the availability of individual genotypes for the MGRB and 45 and Up Study cancer cohorts enabled the direct evaluation of the influence of PS on cancer risk. We first confirmed that our polygenic scoring method estimated individual height using published loci[21]: height PS was significantly predictive of measured height, with a slope of 4.5 cm per PS unit, complete model $R^2 = 0.62$, PS partial $R^2 = 0.14$, $n = 2537$ (Supplementary Fig. 5). We then compared the distribution of PS for prostate, colorectal, and melanoma skin cancer between the 45 and Up

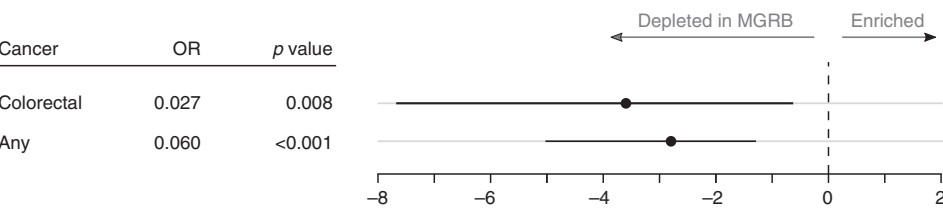

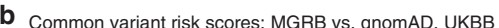

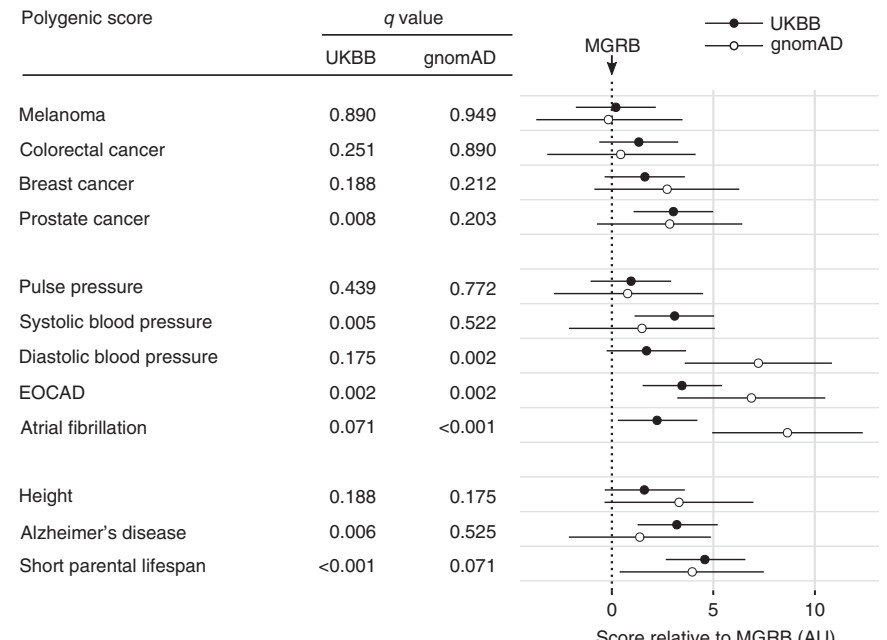

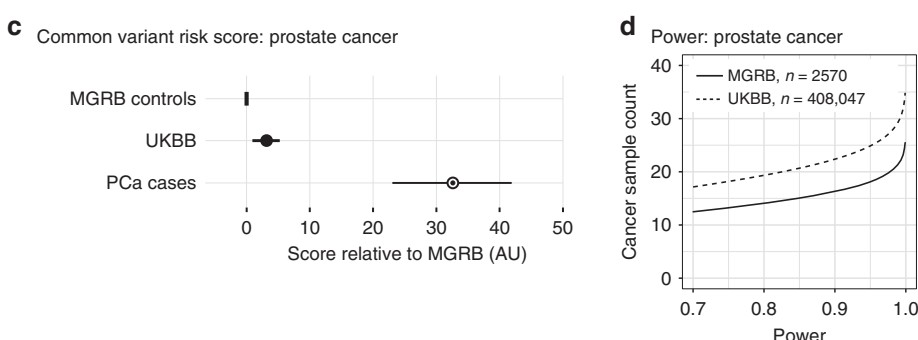

**Fig. 1 The MGRB is depleted for genomic risk relative to reference and disease cohorts. a** The rate of rare pathogenic variants in tumour suppressor genes is lower in MGRB than in a cohort of cancer cases (log odds for an individual to carry a pathogenic tumour suppressor variant shown). **b** The MGRB also has lower polygenic score (PS) estimates for a range of phenotypes, when compared to the gnomAD non-Finnish European population and the UK Biobank samples. MGRB is the reference in **b**, with PS mean set at zero; bootstrap 95% confidence intervals are shown for the difference in PS between MGRB and the reference cohorts (UKBB or gnomAD); higher values indicate a higher polygenic score in UKBB or gnomAD. *q*-Values represent false discovery rate estimates by the Benjamini–Hochberg method[70]. **c** The MGRB has lower PS compared to prostate cancer cases, here considering only samples from the 45 and Up Study. **d** For any given sample size, the MGRB has greater statistical power to detect PS difference from a case cohort than UKBB, demonstrated here for prostate cancer. AU arbitrary units.

cancer-free cases in the MGRB, and individuals from the 45 and Up cohort with these cancers (Supplementary Table 4). Consistent with the relative depletion of rare cancer variants in the MGRB observed above, MGRB individuals had significantly lower polygenic risk scores than cases for prostate cancer (as above, $n = 71$ cases, $n = 278$ MGRB gender- and cohort-matched controls, $t = 3.96$, df = 126, two-sided $p < 0.001$, Fig. 2a)

and colorectal cancer (two-sample *t*-test, $n = 41$ cases, $n = 690$ MGRB controls, $t = 2.46$, df = 44.7, two-sided $p = 0.018$, Fig. 2b), but not melanoma. The contribution of PS to cancer-specific risk was significant: by age 70, individuals with a cancer PS in the top 5% of MGRB had a 7.7-fold increased odds for prostate cancer, and a 3.6-fold increased odds for colorectal cancer, relative to individuals with a score in the bottom 5%.

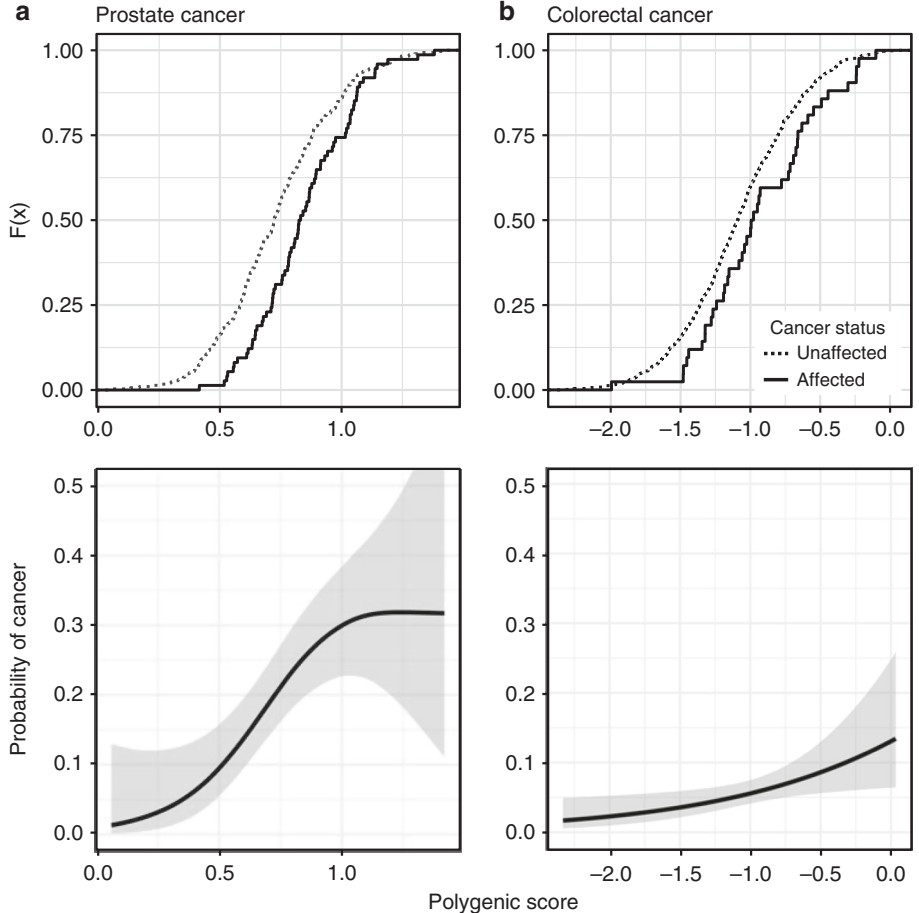

**Fig. 2 Polygenic risk is strongly related to cancer diagnosis.** Cumulative distribution functions (top panels) and associated probability of cancer diagnosis by age 70 (bottom panels) are shown for both prostate cancer (**a**) and colorectal cancer (**b**). Unaffected individuals are MGRB men (prostate), or all MGRB individuals (colorectal) and were completely cancer-free up to age 70; affected individuals were sourced from the 45 and Up Study cancer cohort and had recorded evidence of the relevant cancer diagnosis prior to age 70. Polygenic scores were computed based on reported loci and model coefficients[54,57]. Fits are from logistic regression using a GCV-penalised thin plate spline smooth; bands denote 95% confidence intervals around the mean.

**Clonal somatic variation is detectable by WGS.** In addition to its use as a surrogate for the germline, peripheral blood DNA carries somatic variation reflecting the life history and health state of the donor. Clonal haematopoiesis of Indeterminate Potential (CHIP) occurs in at least 10% of individuals over the age of 65 years, as evidenced by the presence of detectable somatic SNVs in blood DNA[22]. Previous studies have largely used deep whole-exome or targeted sequencing to identify CHIP, but these methods lack sensitivity to detect the copy number variation (CNV) commonly observed in myelodysplasia and leukaemia. Low-depth WGS is a powerful tool for detecting SVs that has been applied to the identification of CHIP-associated SNVs[23], but not CNV. Here we estimate the burden of cancer-associated somatic variation, at both SNV and CNV level, in peripheral blood DNA using whole-genome data in the MGRB cohort.

In total, 184/2570 (7.2%) of MGRB individuals displayed evidence of CHIP, with SNVs associated with overgrowth and neoplasia observed in more than 10% of reads (Supplementary Data 3). Predominantly nonsense mutations (96%), these variants were most commonly seen in *TET2* (47 individuals), *DNMT3A* (23), or *ASXL1* (11) (Supplementary Fig. 6). We also observed known gain-of-function missense variants in *JAK2* V617F (9 individuals), *NRAS* G12D (1), a dominant-negative allele in *DNMT3A*, R882H (1)[24], and a putative loss-of-function variant in *TP53*, C275Y (1). *JAK2* V617F is a recognised driver of myeloproliferative disorders, which are also associated with

*ASXL1* loss[25], and *TET2* and *DNMT3A* loss-of-function variants are frequent in CHIP[5]. In total, the blood of 91/2570 (3.5%) cancer-free MGRB individuals carried deleterious small variation in at least one of these four genes, and 13 individuals had multiple deleterious mutations in this gene set. We next sought evidence for subclonal CNV. Of 2570 MGRB individuals, 1975 were successfully fit to a subclonal CNV model; of these 55 (2.8%) showed evidence of subclonal CNV, as determined by the presence of an aneuploid lineage representing more than 10% of nucleated blood cells (Supplementary Fig. 7). In total, 9.2% (95% CI 7.9–10.5%) of MGRB samples demonstrated evidence of CHIP by either SNV or CNV, consistent with results from deep WES[6]. In sum, subclonal blood DNA changes are detectable from WGS at routine read depths used for germli ne purposes, providing a quantitative fingerprint of age-related somatic events.

**Age-related mitochondrial load is linked to grip strength.** As well as CHIP, ageing is associated with telomere shortening, somatic Y chromosome loss, decreased mitochondrial copy number, and increased mitochondrial heteroplasmy[26–28]. We therefore studied the relationship of age to telomere length, mitochondrial copy number and variation, Y copy number in males, a somatic mutation signature linked to ageing[29], and CHIP. Using standard-depth WGS data from multiple cohorts, consistent patterns of change with age were observed across all six

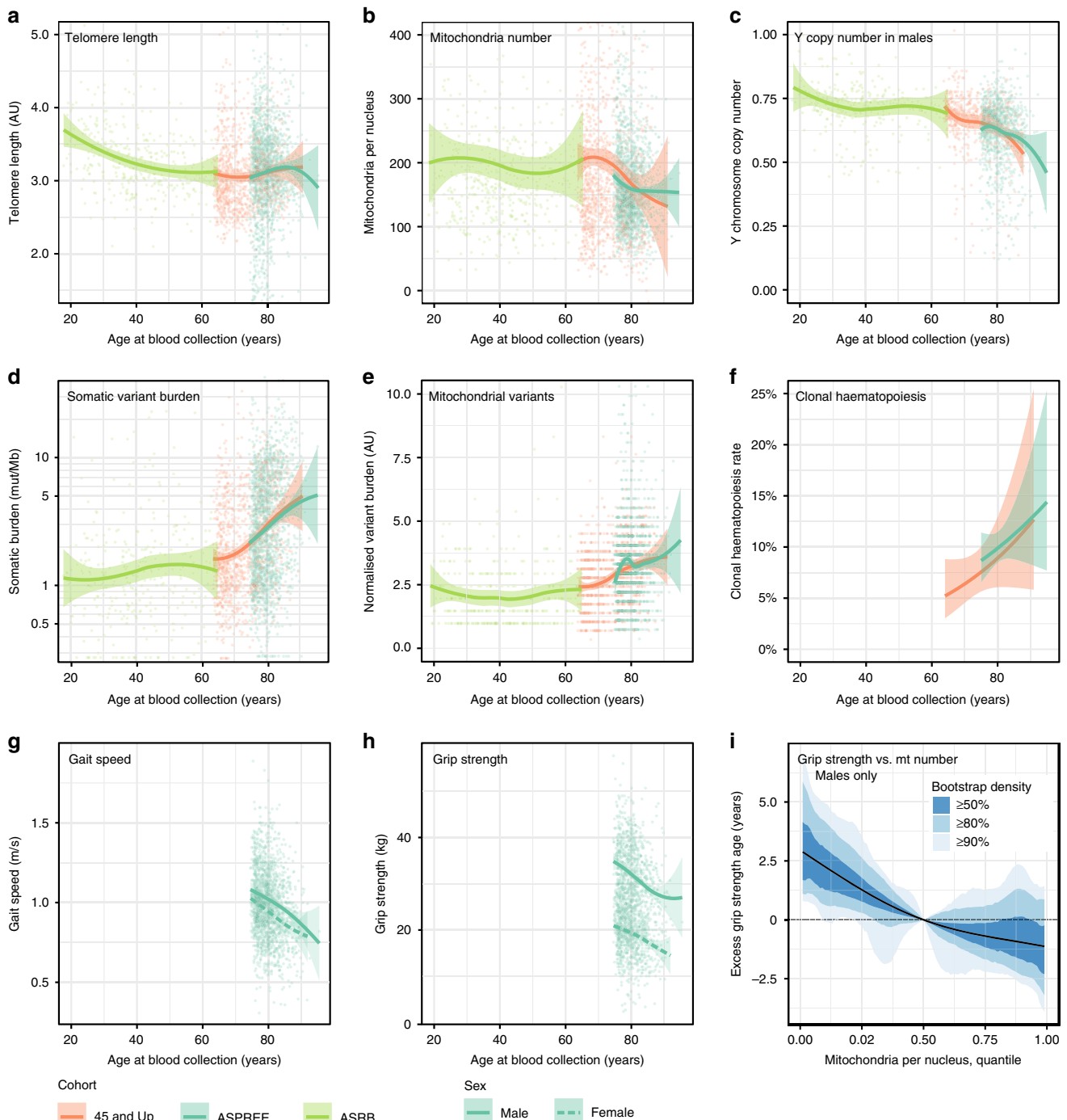

**Fig. 3 Age-related somatic changes are associated with measures of physical function.** Across multiple cohorts, a consistent decrease with age is observed for telomere length (**a**), mitochondria per nucleus (**b**), and Y copy number in males (**c**). In contrast, advanced age is associated with an increase in somatic mutation burden (**d**, **e**) and the fraction of samples with detectable clonal haematopoiesis (**f**), as well as a decrease in the key physical function measures gait speed (**g**) and grip strength (**h**). The count of mitochondria per nucleus is significantly related to grip strength beyond age alone in men, as indicated by the change in effective age as judged by grip strength with varying mitochondria count (**i**). For **a–c**, **g**, **h** individual measurements corrected for cohort batch effect are shown with LOESS smooths, and for **d** a logistic fit was used. Bands around estimates delimit 99% confidence intervals for the mean. Sample numbers were 1853 for the ASPREE cohort, 717 for the 45 and Up Study, and 344 for the ASRB cohort.

somatic metrics (Fig. 3a–f). Compared to a population of younger individuals (the ASRB cohort, median age 40; Supplementary Fig. 8), the MGRB, despite being ascertained on the basis of healthy ageing, was still associated with shorter telomere lengths, increased somatic mutation burden, and decreased Y chromosome and mitochondrial copy number (Table 3). Interestingly, there were differences between each cohort in the relationship

with age, with apparent stabilisation of telomere length in the elderly cohorts past approximately 70 years, compared to the expected progressive shortening with increasing age observed in the younger ASRB cohort. In addition, while mitochondrial copy number/nuclear genome was stable up to age 60, significant declines were observed in the older age groups. The rate of change was significantly different between the young (ASRB) and aged

**Table 3 The rates of somatic measure change with age are different between middle-aged and old individuals.**

| Measure | ASRB | 45 and Up Study | ASPREE |
|---|---|---|---|
| Individuals | 344 | 717 | 1853 |
| Percent female | ND | 59.3% | 48.2% |
| Median age (range) | 40 (18–65) | 70 (64–1) | 79 (75–95) |
| Telomere length (AU/decade) | **−0.115** [−0.157, −0.073] | 0.040 [−0.010, 0.090] | **0.115** [0.035, 0.196] |
| Mitochondria count ($\log_{10}$ mt/nucleus/decade) | −0.004 [−0.018, 0.010] | **−0.046** [−0.065, −0.027] | **−0.038** [−0.059, −0.017] |
| Y copy number in males (Y chromosomes/nucleus/decade) | −0.011 [−0.022, 0.001] | **−0.050** [−0.068, −0.033] | **−0.043** [−0.065, −0.021] |
| Somatic variant burden ($\log_{10}$ variants/Mb/decade) | 0.038 [−0.002, 0.079] | **0.207** [0.167, 0.247] | **0.228** [0.173, 0.282] |
| Mitochondrial variants (mt variants/decade) | 0.051 [−0.177, 0.278] | **1.665** [1.315, 2.015] | **0.893** [0.195, 1.591] |

Numbers show the rate of change of each somatic measure with age in the middle-aged ASRB cohort (median age 40), and the older MGRB cohorts (median age 70 or older). Changes are significantly different between the younger ASRB and older MGRB cohorts, and consistent within the two older MGRB cohorts. Linear model slopes as change per decade are reported for each of five somatic measures in each cohort, with 95% Wald confidence intervals. Values significantly different from zero are represented in bold. Note that somatic burden and mitochondrial count per nucleus are reported on the natural logarithm scale. ND, not determined due to data use agreement constraints

(MGRB) cohorts (5 likelihood ratio tests on linear fits, Holm correction, all $p < 0.003$), while the rate of change of the two aged cohorts was consistent across all measures (5 likelihood ratio tests, Holm correction, all $p > 0.28$). Taken together, these results are suggestive of altered kinetics of age-related somatic mutation in the elderly compared to younger populations, although we note longitudinal measurements will be necessary to definitively establish this relationship.

We next considered whether individual age-related genomic measures may reflect physical function status, independently of chronologic age. To address this question, somatic changes in MGRB samples from the ASPREE cohort were studied in the context of age, grip strength, and gait speed, all representing key predictors of age-related morbidity[30]. As expected, grip strength and gait speed both consistently decreased with age in both genders (Fig. 3g, h). To correct for the strong influence of age on all measures, a conditional analysis was performed to explore whether any somatic measures were associated with physical function even when age is taken into account. Intriguingly, we found that grip strength was positively correlated with the count of mitochondria per nuclear genome, but only in males (two-stage bootstrap test, first stage $p = 0.051$, validation $p = 0.036$; Supplementary Table 5).

To illustrate the magnitude of effect of mtDNA copy number on grip strength in men, we modelled the change in "effective age" as determined by grip strength, as a function of mtDNA copy number. This revealed that men with an mtDNA copy number in the lowest 5% for their age have the same grip strength as men with average mtDNA levels, but who are 2.5 years older (Fig. 3i).

## Discussion

Understanding the genetic underpinnings of healthy ageing is as important as, and relevant to, understanding the genetic basis of disease. The next decade will see the fruits of population-scale sequencing programmes, much of which will be aimed at understanding the genetic origins of disease. To realise this mission, we need to understand the spectrum of genetic variability in the healthy, and whole-genome datasets of healthy controls will be essential to identify genetic variation unrelated to disease[31]. To this end we created the MGRB, a WGS resource of deeply phenotyped aged individuals[11]. This report describes the first data release of the MGRB comprising 2570 individuals; the MGRB will grow in time to contain sequence and phenotype data of 4000 healthy elderly Australians.

Although depletion of some common disease-related alleles has been reported in the healthy aged[4,32], the MGRB reveals a general depletion in disease-associated common and rare variation, relative to both affected cases, as well as datasets frequently used as

controls in genetic studies, but not specifically depleted for disease phenotypes. In addition, the MGRB was enriched for protective alleles linked to healthy ageing. Our data also substantiate the premise that extreme phenotype enrichment can enhance statistical power in case:control genetic studies[10] (Fig. 1c, d).

Despite being healthy, over 1% of the MGRB still carry pathogenic small variants that are clinically reportable under current ACMG guidelines (Table 2), consistent with previous observations in the healthy elderly[4]. A detailed review of individual phenotypes from a subset of mutation carriers excluded even subclinical manifestations of the expected disorders. These data suggest that many apparently pathogenic small variants have variable penetrance, echoing a theme emerging from population genomic studies. Additionally, several rare SVs were identified that may abrogate function of clinically reportable genes (Supplementary Table 3). Future studies using whole-genome-based data will benefit from the MGRB in quantitating the contribution of structural variation to ageing and disease. These observations suggest the MGRB may provide a filter for rare variants currently thought pathogenic in a clinical context.

The rate of pathogenic variation observed in the MGRB appeared lower than that previously reported in European cohorts that were not selected on the basis of healthy ageing[33–35]. However, the comparison of rare variation rates between cohorts is affected to a greater extent than common variants by inter-platform variation and subtle population stratification. Further study will be required to definitively test whether the well elderly have a significantly lower burden of pathogenic rare variation to the general population.

The ageing process is accompanied by the emergence of somatic mutations in tissues other than blood, mitochondrial depletion and heteroplasmy, and progressive telomere shortening[5,6,26–28,36]. We developed a suite of methods to detect these age-related changes using 30X WGS, and applied it to the elderly MGRB and a younger cohort. Telomere shortening itself may directly increase the likelihood of neoplasia[37], while oncogenic mutations in genes such as TP53 may rescue the effects of telomere loss[38]. Telomere dysfunction has been associated with impaired mitochondrial function[39], linking these genomic features of ageing. Many of the somatic changes of ageing observed in the MGRB are consistent with marrow stem cell depletion, as observed in telomerase-deficient mice[40].

Interestingly, we observed a shift in the age trajectory of multiple somatic metrics in the elderly compared to younger individuals, coincident with the emergence of clonal hematopoiesis. It is paradoxical that, in this and other cohorts, somatic clonal expansion driven by oncogenic mutations appears compatible with normal organ function[5,6,36]. It is even possible that neoplastic events, such as telomere stabilisation, loss of tumour

suppressor genes, or acquisition of oncogenic kinase mutations, might increase clonogenic efficiency of an ageing marrow stem cell compartment. Some support for this concept, reminiscent of antagonistic pleiotropy, comes from mice carrying a hypermorphic form of *Trp53*, in which protection from neoplasia was accompanied by accelerated hematopoietic ageing and diminished marrow reserve[41]. If true, these findings suggest that strategies that suppress tumour formation may accelerate ageing.

We observed an intriguing link between somatic burden and decline in physical function, providing a potential measure of what distinguishes individuals sharing the same age, but different physical function status. The relative depletion of mitochondria per leucocyte appeared to be associated with reduced grip strength in males, after adjustment for age. This finding is consistent with evidence that mitochondrial dynamics are strongly involved in ageing and function, particularly in males[26]. We note that our power to detect such an effect is low when using a well elderly cohort, but believe there will be great interest in deriving quantitative measures of biological ageing from standard-depth WGS.

Although the largest cohort of healthy elderly whole genomes amassed to date, the MGRB is still subject to limitations as a research and clinical tool. The investigation of extremely rare variants is limited by the MGRB's size, and complicated by batch effects in rare variant calls. Furthermore, the MGRB comprises almost exclusively white Australians, and follow-up studies will be required to assess the spectrum of genetic variation in the healthy elderly from other backgrounds. The MGRB was recruited on the basis of a restricted definition of healthy ageing, being depletion of cancer, cardiovascular disease, and dementia, and MGRB participants do bear other morbidities. However we note that the deep phenotype which accompanies the MGRB enables more focussed participant selection and the construction of for-purpose subset cohorts, making the MGRB of value as a universal control that can be depleted of any measured phenotype. Finally, although we observed associations between somatic measures and age that are suggestive of changes in ageing kinetics, this cannot be definitively established using our cross-sectional study design. Further studies with longitudinal samples will be required to verify our hypothesis of altered ageing kinetics, for which the methodology established here will be valuable.

Quantitative biomarkers of age may provide a summative metric of diverse genetic and environmental effects on health. Interpreted as endophenotypes, such biomarkers show promise to increase our ability to detect genetic patterns associated with ageing rate, but their true utility may be greater still as clinical tools in their own right. By encoding the aggregate influence of complex and potentially unmeasurable genetic and environmental effects over the life of an individual, biomarkers of age may represent health and disease risk with greater fidelity than external indicators such as calendar age or functional state.

Particularly with respect to cancer, the DNA-based measures of biological age we have demonstrated here may represent an individual's underlying mutation rate, and therefore true cancer risk, due to combined genetic and environmental factors. This biomarker-centric perspective on cancer risk represents a synthesis and simplification of the traditional genotype- and environment-centric views, and we believe is a promising lens through which to consider disease risk, and differentiate normal compensatory changes associated with ageing, from those that precede malignancy.

## Methods

**Experimental model and subjects**. Participants of the MGRB were consented through the biobank programmes of the ASPREE and 45 and Up studies[11,13]. At the time of blood collection, each participant was aged 60 years or older.

Samples from the ASPREE study were from individuals aged 75 years or older at the time of enrolment, with no reported history of any cancer type, no clinical diagnosis of atrial fibrillation, no serious illness likely to cause death within the next 5 years (as assessed by general practitioner), no current or recurrent condition with a high risk of major bleeding, no anaemia (haemoglobin > 12 g/dL males, >11 g/dL females), no current continuous use of other antiplatelet drug or anticoagulant, no systolic blood pressure ≥180 mm Hg and/or a diastolic blood pressure ≥105 mm Hg, no history of dementia or a Modified Mini-Mental State Examination (3MS) score ≤77, and no severe difficulty or an inability to perform any one of the 6 Katz basic activities of daily living.

Samples from the 45 and Up Study were from individuals with no self-reported history of cancer, heart disease, or stroke. Neurological disease was not explicitly excluded, but participants were required to correctly self-complete a health survey at enrolment. We confirmed no record of cancer diagnosis in the NSW Central Cancer Registry, and no record of cancer diagnosis in the NSW Admitted Patient Data Collection, for all 45 and Up Study individuals in the MGRB.

Participants in the Australian Schizophrenia Research Bank (ASRB) were recruited through a national media campaign and consented to data and sample collection and genomic analyses following discussion with a clinical assessment officer[42]. UK Biobank samples were sourced from the UK Biobank Resource under Application Number 17984.

**Ethics**. The ASPREE Biobank study was approved by the Monash University Human Research Ethics Committee, and subsequent WGS of Australian ASPREE participants was approved by the Alfred Hospital Ethics Committee. The use of 45 and Up Study samples in the MGRB is covered by ethics approvals from the University of New South Wales Human Research Ethics Committee and the NSW Population & Health Services Research Ethics Committee. The use of the ASRB data was approved by the University of Newcastle Human Ethics Research Committee.

**Sample collection and processing**. For ASPREE participants of the MGRB, peripheral blood samples were processed to buffy coat within 4 h of collection, then stored at −80 °C. DNA was later purified from buffy coat via magnetic bead extraction (Qiagen).

For 45 and Up Study participants of the MGRB, peripheral blood samples were refrigerated at 4 °C and processed to buffy coat within 48 h of collection. Buffy coat was stored at −80 °C, and DNA purified via column extraction (Qiagen).

ASRB participant PBMCs were extracted from whole blood by centrifugation in Lymphoprep (Vital Diagnostics). Genomic DNA (gDNA) was extracted from PBMCs using salt extraction and quantified by PicoGreen assay (Life Technologies). The integrity of gDNA was determined by agarose gel electrophoresis prior to sequencing.

**Sequencing**. WGS of the MGRB, 45 and Up cancer, and ASRB cohorts was performed on Illumina HiSeq X sequencers at the Kinghorn Centre for Clinical Genomics (KCCG), Sydney, using paired-end Illumina TruSeq Nano DNA HT libraries and v2.5 clustering and sequencing reagents. Each sample was sequenced on one HiSeq X lane.

**Sequence alignment and processing**. All sequence data generated at the KCCG were processed following the Genome Analysis Toolkit (GATK) best practices[43]. We first defined a custom reference genome tailored to Illumina HiSeq X sequencers, being the 1000 Genomes Phase 3 decoyed version of build 37 of the human genome, with an added contig of NC_001422.1 to act as a decoy for the HiSeq-specific ΦX174 sequence spike-in. Reads were aligned to this reference using bwa 0.7.15 mem in paired mode, and duplicates marked with biobambam2 2.0.65 bamsormadup, with a minimum optical pixel distance of 2500. All other parameters for both bwa and bamsormadup were left at defaults. For high-depth samples run on multiple sequencing lanes, data merging was performed at this point using samtools 1.5. Indel realignment and base quality score recalibration of mapped reads were performed using GATK 3.7-0 and best practices parameters; unmapped reads were left unmodified. GATK HaplotypeCaller was used to generate g.vcfs from all single-lane realigned and recalibrated BAMs using recommended parameters. Pipeline steps were accelerated using GNU parallel 20170722 (ref. [44]).

**Locus confidence tiers**. We defined locus confidence tiers for WGS genotyping on the basis of prior annotations, sequence complexity, and empirical metrics on our data. Locus tiers ranged from 1 to 3, with 1 indicating the highest confidence in WGS variant detection performance and 3 the lowest.

To specify the locus confidence tiers, we first identified regions of the genome which empirically had unusual coverage in the MGRB and 45 and Up cancer sequencing data. For each sample we defined bounds on the expected sequence coverage as the 0.001 and 0.999 quantiles of a Poisson distribution, with rate equal to the modal nonzero coverage observed across all autosomal loci within that sample. As typically 15 reads are required for high genotyping performance[45], the lower bound was thresholded to always be at least 15. Within each sample, we defined each autosomal locus as being either in-bound (depth within the sample-

specific bounds) or out-of-bound. We then calculated across all samples the rate at which each locus was out-of-bounds, considering the entire MGRB cohort. Regions for which this rate exceeded 5% (in other words, loci which had unusual coverage in at least 5% of MGRB + 45 and Up cancer samples) were marked as problematic. These problematic regions were smoothed by a morphological closing operation followed by an open operation, with structuring elements being centred intervals on the genome of size 131 and 11 bp, respectively, to yield a final definition of regions of unusual depth in the MGRB cohort. These regions totalled 409 Mb, 13.0% of the reference genome, 13.2% of the canonical chromosomes (1–22, X, Y), and 14.9% of the CCDS coding sequence (accessed 21 Nov 2017).

We then defined a poor-quality subset of the genome as all loci within 5 bp of the union of: the unusual depth regions, repeat regions identified by RepeatMasker, low complexity regions of the reference genome detected by mdust with default parameters, excludable regions from the ENCODE project, and poorly aligned or non-unique regions from the ENCODE project (Supplementary Data 4). This poor-quality subset totalled 1832 Mb in size, 58.4% of the reference genome, 59.0% of the canonical chromosomes, and 18.1% of the CCDS coding sequence.

Variants in non-canonical chromosomes, the pseudoautosomal regions (X: 60001–2699520, 154931044–155260560; Y: 10001–2649520, 59034050–59363566), or within the poor-quality subset of the genome defined above, were assigned to the lowest confidence tier 3. For the remaining variants in canonical chromosomes, if the variant overlapped a high-confidence HG001 region identified by the GiaB consortium v3.3.2 (ref. [14]) it was assigned the highest confidence tier 1, else it was assigned an intermediate confidence tier of 2. In total, 81.9% of the CCDS coding genome was in confidence tier 1 or 2 (Table 4).

**Initial sample quality control.** Poor-quality MGRB and 45 and Up cancer samples were identified on the basis of genotype metrics at a small diagnostic set of loci. All 3033 single-lane samples were genotyped at SNP loci on the Illumina Infinium QC Array 24 v1.0, using GATK GenotypeGVCFs, and quality metrics calculated within Hail v0.1 (ref. [46]). A total of 2904/3033 (95.7%) samples passed initial quality thresholds (Table 5). Of these, 14 (0.5%) had a reported sex that did not match their genetic sex, as determined from the X chromosome inbreeding coefficient; these sex-discordant samples were not considered further. In total, 2890/3033 (95.3%) MGRB and 45 and Up cancer samples passed initial quality control (QC).

**Small variant genotyping and final QC.** The 2890 MGRB and 45 and Up cancer samples passing initial QC were joint called in a single batch using GATK GenotypeGVCFs, and imported to Hail v0.1 for processing. A second round of QC (Table 5) identified an additional 31 samples with poor-quality metrics not revealed by the initial QC round; these were dropped. The PCRELATE component of the GENESIS 2.8.0 package[47] was used to determine structure-corrected relatedness between the 2859 samples remaining, using autosomal SNPs LD-pruned with an $r^2$ threshold of 0.1, KING robust relatedness estimates from SNPrelate 1.12.1 (ref. [48]), and without a population reference cohort. Fourteen pairs of individuals related to second degree or closer were identified and excluded from the cohort. MGRB (cancer-free) and 45 and Up cancer samples were split into separate

cohorts at this point, and four 45 and Up cancer samples excluded on the basis of incomplete or inconsistent clinical data. In summary 2841 unrelated samples passed all data quality requirements, comprising 2570 cancer-free MGRB individuals, 269 45 and Up cancer samples, and the reference materials RM 8391 and RM 8398.

**ASRB processing and quality control.** Sequence processing and quality control for the ASRB cohort proceeded as described for the MGRB. 344/476 ASRB samples passed all QC thresholds and were used for subsequent analysis.

**Cohort population structure.** The MGRB cohort population structure was determined using principal components analysis (PCA), with reference to the 1000 genomes (1000 G) populations. A merged dataset of all MGRB and 45 and Up cancer genotypes and the 1000G Phase 3 genotypes was generated in Hail. To ensure high genotype concordance between platforms, merged variants were restricted to autosomal strand-specific SNPs in Tier 1 regions of the genome (see Locus confidence tiers), with a 1000G allele frequency in the range of 5–95%, and no evidence of deviation from Hardy–Weinberg equilibrium within any of 17 homogeneous 1000G populations ($P_{HWE} > 0.01/17$ for each of population codes BEB, CDX, CEU, CHB, CHS, FIN, GBR, GWD, IBS, ITU, JPT, KHV, LWK, MSL, STU, TSI, and YRI). Merged variants were LD-pruned in Hail with an $r^2$ threshold of 0.1, and PCA performed in Hail on biallelic variants with a combined MGRB and 1000G allele frequency in the range of 5–95%.

A hierarchical eigenvalue decomposition discriminant analysis classifier was constructed to assign MGRB samples to 1000G populations on the basis of PCA scores. The first classifier layer predicted a sample's 1000G superpopulation (AFR, AMR, EAS, EUR, or SAS), and the second a sample's European population (CEU, FIN, GBR, IBS, or TSI), conditional on EUR being the predicted superpopulation by the first layer. Models were trained on 1000G sample scores only using PC1–4 as predictors, and then were applied to predict population source for the MGRB samples. All models were implemented using mclust v5.3 (ref. [49]).

**Small variant processing and annotation.** Small variant processing and annotation was performed within Hail v0.1. Variant consequences were determined using Ensembl VEP 90 with default Ensembl release 90 databases. Variants were further annotated with 1000 genomes Phase 3, Haplotype reference consortium, GnomAD, and dbSNP allele frequencies, as well as ClinVar, CATO, and Eigen annotations (see Supplementary Data 4 for resource versions).

**Germline SV detection.** Germline SVs in the MGRB and 45 and Up cancer cohorts were detected using GRIDSS v1.4.1 (ref. [50]), excluding regions in the Encode DAC Mappability Consensus Excludable list (Supplementary Data 4). Where possible, linked sets of breakend calls resulting from a single rearrangement were merged into higher-level structural events. To eliminate overlap with GATK indel calls and enable assessment of cohort frequencies, SV events were filtered to be of length at least 50 bp, and those of the same type within a window of 100 bp were merged to the one call.

Germline MEIs were identified using Mobster v0.2.2 (ref. [51]) without blacklisting existing mobile element regions. MEI calls were then processed to remove false-positive events in existing mobile element regions and to estimate variant zygosity by local realignment to the reference genome. MEIs occurring in different samples within 100 bp of each other were merged to the one call.

**Rare variant burden comparison.** To compare rare variant burden between the platform-matched MGRB and 45 and Up cancer cohorts, missense or nonsense variants (as judged by VEP) in ACMG SF 2.0 cancer-associated genes were joint called across both cohorts, and each variant scored for pathogenicity by ACMG criteria, blinded to cohort. The rate of individuals carrying pathogenic variants was then directly compared by Fisher's test. To exclude potential confounding due to source cohort, the 45 and Up component of the MGRB only was compared to the 45 and Up cancer cases.

**Table 4 Quantity in megabases of genomic regions in each locus confidence tier.**

| Locus confidence tier | Reference genome | Canonical chromosomes | CCDS |
|---|---|---|---|
| 1—highest | 1212 | 1212 | 25.40 |
| 2 | 52 | 52 | 1.19 |
| 3—lowest | 1874 | 1832 | 5.88 |
| Total | 3137 | 3096 | 32.47 |

Canonical chromosomes are 1–22, X, and Y; CCDS represents the Consensus CDS set

**Table 5 Quality metric conditions for samples to pass quality control.**

| Metric | Initial QC (Infinium SNPs) | Final QC (full genotypes) |
|---|---|---|
| Call rate | >0.98 | >0.98 |
| Depth standard deviation | <10 | <10 |
| VAF standard deviation at loci called heterozygous | <1 | <1 |
| Heterozygous:homozygous variant ratio | <2 | <2 |
| X chromosome inbreeding coefficient | <0.2 or >0.8 | Not tested |
| Singleton rate | <0.001 | Not tested |

Two rounds of quality control (QC) were performed, with different metric cutoffs: a first round based on genotypes at Illumina Infinium QC Array 24 SNPs only and a second round based on genotypes called across the whole-genome. Only samples passing all cutoffs in both rounds were included in the MGRB Phase 2 release

**Genome-wide common variant frequency comparison.** To compare patterns of common variation between the MGRB and other cohorts, we merged the MGRB variants with gnomAD v2.0.1 non-Finnish European (NFE) WGS allele frequencies, and allele frequencies from a subset of the UK Biobank genotype set, consistent of participants who self-identified as white British and who had confirmed European ancestry by PCA[52]. To minimise the influence of technical artefacts, variants were restricted to strand-specific biallelic SNPs listed in the EBI GWAS database, that were located in regions of the genome covered by the Genome in a Bottle standard, and were sequenced to a depth of at least 15 in at least 98% of samples in both the MGRB and gnomAD WGS cohorts. Further, variants which were not observed in one or more cohorts, or were genotyped at a rate of less than 97% in any cohort, were excluded. In total, 21,033 SNPs remained following this filtering, with very similar allele frequencies across all cohorts (Supplementary Data 2, sheet 1; Supplementary Fig. 3); these loci and frequencies were used in the following common variant analyses.

We tested for phenotype-linked bias in allele frequency between the cohorts as follows. For a given phenotype-associated set of variants, each variant was scored on two metrics: its variant allele frequency enrichment or depletion in MGRB vs gnomAD or UKBB, and the positive or negative association of the variant allele with the trait. A Fisher's exact test was then used to test for dependence of variant enriched/depleted status on the trait direction of effect, with deviation from the null indicating an allele frequency bias between MGRB and gnomAD or UKBB that is specific to the phenotype considered.

Three sets of variants were tested by this procedure: a test set of variants reported to be associated with phenotypes depleted in the MGRB (Supplementary Data 2, sheets 2–3), and two negative control sets of variants linked to anthropometric traits (Supplementary Data 2, sheets 4–5), or behavioural traits (Supplementary Data 2, sheets 6-7).

**PS estimation and testing.** PSs were calculated as $S_i = \sum_j \beta_j d_{ij}$ where $S_i$ is the PS for individual $i$, $\beta_j$ the GWAS-reported coefficient for a single variant allele at locus $j$, and $d_{ij}$ is the variant allele dosage for individual $i$ at locus $j$. We considered only autosomal variants, and if a variant dosage was not available for an individual, it was imputed as $\hat{d}_{ij} = 2f_j$, with $f_j$ the variant allele frequency reported by the source publication. To reduce bias due to this imputation, variants with a call rate under 97% were excluded from PS calculation in all individuals.

PS GWAS coefficients were derived from each source publication by the following procedure. Firstly, all loci that were reported to be genome-wide significant in replication were selected, along with the effect allele's association coefficient in the replication cohort. If replication results were not reported, the derivation cohort loci and coefficients were used. Regression coefficients for binary traits were then transformed to a log-odds scale; coefficients for the continuous traits of height and parental lifespan were used as-is. Loci were converted to GRCh37-centric coordinates, and where possible any loci falling within tier 3 regions (see section Locus confidence tiers) were imputed to an alternative locus falling in a tier 1 or 2 region, chosen so that the original and alternative loci had an $R^2$ of at least 0.9 as assessed in the GBR population of 1000 genomes by NCI LDlink[53]. If no alternative loci fulfilled this criterion, the original locus was dropped from the PS. PSs were derived by this procedure for colorectal cancer[54], melanoma[55], breast cancer[56], prostate cancer[57], blood pressure[58], early-onset coronary artery disease[59], atrial fibrillation[60], height[21], Alzheimer's disease[61], and parental longevity[1]. The Alzheimer's disease PRS as originally reported lacked the highly significant *APOE* locus; accordingly this locus was manually added to the PRS using the tag SNP rs10414043, and an estimated $\beta = 1.34$ (ref. [62]). rs10414043 was used in preference to the more conventional rs429358 as the latter was not robustly genotyped on all platforms. All loci, alleles, and coefficients used in the PRS calculations are detailed in Supplementary Data 2, sheet 8.

An approximate bootstrap procedure was used to test for PS shift between MGRB, gnomAD, UKBB, and the 45 and Up cancer cohort, accounting for genetic drift between populations (Supplementary Fig. 9). All cohorts were first collapsed to allele-frequency data only, with individual genotypes discarded. PS variants were subset to those called at a rate of at least 97% all cohorts, and with an absolute difference in alternate allele frequency between MGRB, gnomAD, or UKBB of less than 4%. To include the effect of genetic drift in the bootstrap, the allele frequency differences between MGRB, and each of the gnomAD, UKBB, and 45 and Up cancer cohorts were calculated as $d_{2,i} = \left( f_{MGRB,i} - f_{Other,i} \right) \div \sqrt{f_{Other,i}\left(1 - f_{Other,i}\right)}$, where $f_{Cohort,i}$ denotes the variant allele frequency at locus $i$ in cohort Cohort. $d_{2,i}$ was calculated for all well-genotyped loci (not just those in PSs). $d_{2,i}$ is related to the fixation index $F_{ST}$, and the distribution of $d_{2,i}$ values between a cohort and MGRB reflects the genetic distance between these groups of individuals. For a given PS and comparison cohort Other, testing then proceeded as follows. A bootstrap Australian reference cohort (ARC) was generated by shifting the Other allele frequencies based on $d_2$:

$$f_{ARC,i} = f_{Other,i} + d_2^{(i)}\sqrt{f_{Other,i}\left(1 - f_{Other,i}\right)}$$, where $d_2^{(i)}$ has been sampled with

replacement from $d_2$. This simulates an unseen comparison cohort ARC which has the same mean genetic distance from MGRB as the comparison cohort Other, but which we would expect has no allelic shift due to the phenotypic depletion that we hypothesise is present in MGRB. To test this hypothesis we calculate the expected PS

in both MGRB and in ARC, as $s_{Cohort} = \frac{2}{n}\sum_{j=1}^{n} f_{Cohort,j}\beta_j$, where $j = 1...n$ indexes the PS loci, and estimate the MGRB score depletion as $s_{MGRB} - s_{Other}$. This depletion statistic was calculated for each comparator cohort and PS, for 100,000 bootstrap rounds.

To facilitate comparisons between scores of different scales, bootstrap distributions for each score were normalised so that the $s_{UKBB}$ score had a mean of zero and a standard deviation of 1. The 95% bootstrap confidence intervals for each cohort were then defined as the 0.025 and 0.975 quantiles of the normalised scores. Approximate two-sided $p$ values for the MGRB score depletion were calculated as $p \approx \frac{2}{B+1}\left(\frac{1}{2} + \min\left(\sum_{k=1}^{B}\left[s_{MGRB,(k)} < s_{Other,(k)}\right], \sum_{k=1}^{B}\left[s_{MGRB,(k)} > s_{Other,(k)}\right]\right)\right)$ where $s_{MGRB,(k)}$ and $s_{Other,(k)}$ are samples from the bootstrap mean values for cohorts MGRB and Other, respectively, $B = 100,000$ is the number of bootstrap rounds, and [] denotes Iverson brackets.

The statistical power improvement from using the MGRB extreme phenotype cohort as a control vs gnomAD was estimated by asymptotic approximation. Bootstrap distribution means of the mean prostate cancer score difference between the 45 and Up cancer cases, and gnomAD and MGRB controls were used as mean shift values for statistical power calculation. After correcting for varying cohort sample size, bootstrap distribution variance was highly consistent across all three cohorts, and pooled variance scaled to a sample size of 1 was used as the dispersion parameter. Power was then calculated across a range of sample sizes for both the MGRB vs 45 and Up cancer, and gnomAD vs 45 and Up cancer tests, by direct root finding of the relevant $t$ distributions. Finally, the power vs sample size relationship was inverted by piecewise linear interpolation to yield the sample size vs power curves.

Individual genotypes were available for both MGRB and 45 and Up cancer cohorts. In these cases, a secondary analysis was performed that directly compared the distributions of individual PSs between cohorts. Height prediction was validated by ordinary linear regression of measured individual height against the polygenic height predictor[21] with additional additive linear covariates of sex and age at measurement; no evidence for model misspecification was observed. The association between PS and risk of specific cancers was assessed by logistic regression, with the effect of PS on cancer risk modelled by GCV-penalised thin plate splines. Comparisons were restricted to the specific cancers of prostate, colorectal, and melanoma, as other cancers were either poorly sampled in the 45 and Up cancer cases, or did not have PSs defined.

**Incidental somatic variant detection.** Somatic variants were identified in post-BQSR BAM files using FreeBayes, with options: --pooled-continuous --standard-filters --min-alternate-fraction 0 --min-alternate-count 3 --hwe-priors-off --allele-balance-priors-off --use-mapping-quality. FreeBayes was restricted to detecting variants within 10 kb of RefSeq genes in the COSMIC Cancer Gene Census downloaded 11 December 2017. Variant annotation was performed using the Ensembl VEP[63] release 90, with default options, and variants were notated with COSMIC 83 frequencies.

Annotated variants were filtered to retain only non-synonymous variation (missense, splice donor or acceptor, start lost, stop gained, frameshift, or inframe indel) affecting Cancer Gene Census Tier 1 genes, with a maximum population allele frequency of less than 0.1%, a variant allele fraction (VAF) of at least 10%, and three or more reads supporting the variant. We then identified likely driver mutations from these filtered variants by the following criteria: either a variant had a HIGH consequence in a canonical tumour suppressor gene transcript or the variant was observed at least 100 times in the COSMIC database. Consequences and canonical transcripts were as defined by Ensembl VEP; tumour suppressor genes were Tier 1 genes from the COSMIC Cancer Gene Census with a TSG annotation.

**Telomere length.** Telomere lengths were estimated using Telseq v0.0.1 (ref. [64]). To reduce batch effects between the ASRB and MGRB cohorts, ASRB telomere length estimates were calibrated using Deming regression, fit to 85 ASRB samples sequenced both in the original ASRB batch, and contemporaneously with the MGRB.

Telomere length estimation by Telseq was validated by qPCR on a subset of 120 samples from the ASRB and MGRB cohorts (Supplementary Fig. 10)[65]. Briefly, qPCR was conducted in triplicate. Reactions included genomic DNA (5 ng), 2× Rotor-Gene SYBR Green Master Mix (Qiagen), 500 nM Tel forward [5′-CGGTTT(GTTTGG)_5GTT-3′] and 500 nM Tel reverse [5′-GGCTTG(CCTTAC)_5CCT-3′] or 300 nM 36B4 forward [5′-CAGCAAGTGGGAAGGTGTAATCC-3′] and 500 nM 36B4 reverse [5′-CCCATTCTATCATCAACGGGTACAA-3′] in a 25 μL reaction. Amplification was conducted in a Rotor-Gene Q qPCR cycler (Qiagen) at 95 °C for 5 min, followed by 30 cycles of 95 °C for 7 s and 58 °C for 10 s (telomere reaction) or 35 cycles of 95 °C for 15 s and 58 °C for 30 s (single copy gene reaction). Telomere content for each sample was determined by the telomere to single copy gene ratio (T/S ratio) by calculating $\Delta C_t = C_{t_{telomere}} \div C_{t_{single\,copy\,gene}}$. The T/S ratio of each sample was normalised to the mean T/S ratio of a reference sample, which was included in each run. The experiment was accepted if the reference sample T/S ratio ranged within 95% variation interval, and if the standard curve had a high correlation factor ($R^2 > 0.95$).

**Mitochondria and Y chromosome copy number**. Mean mitochondrial genome copy number in each sample was estimated using read counts, as $2 \times (R_{MT} \div S_{MT}) \div (R_A \div S_A)$, where $R_Z$ and $S_Z$ denote the number of reads mapping to contig set $Z$ and the total size of contig set $Z$, and MT and A denote mitochondrial and autosomal contigs, respectively. Read counts were mapped and aligned reads reported by samtools idxstats, and were not corrected for read duplication. Patch contigs were not included in counts. Y copy number in males was estimated by an analogous procedure, as $2 \times (R_Y \div S_Y) \div (R_A \div S_A)$.

**Mitochondrial variants**. Variants in the mitochondrial genome were detected using FreeBayes, considering only reads with base quality over 24 and mapping quality over 30; all other parameters were left at defaults. Variants with fewer than 10 alternate reads, or an alternate allele fraction under 0.001, were discarded. For each variant passing these filters a Phred-like quality score $q$ was calculated as $q = -10\log_{10}(1 - F(n; p, N))$, with $n$ the count of alternate allele reads, $N$ the total depth at the variant locus, $p = 0.0025$ a fixed error rate estimate, and $F(n; p, N)$ the cumulative density function of a binomial distribution with $N$ draws and success probability $p$. Variants with $q < 30$, high-depth variants ($n > 15$) with an alternate read strand bias of greater than 0.9, or variants in the highly variable locations MT:302–319 or MT:3105–3109 were discarded. The final metric of mitochondrial variant burden for a sample was defined as the number of low-frequency (variant allele fraction under 0.01) variants passing all above filters in that sample.

**Somatic single-nucleotide variants**. Somatic SNV burden was estimated using a combination of statistical filtering and spectral denoising. Putative somatic SNVs were first identified on the basis of a variant allele frequency that was statistically inconsistent with either machine error or germline variation. The burden of these variants in each sample was then dimensionally reduced by spectral factorisation, and per-sample signature scores used as the final somatic variant estimates.

We first developed a statistical filtering procedure to identify likely somatic variants that uses dynamic thresholds to optimise sensitivity while controlling signal to noise ratio. This procedure calls a variant at a given locus as likely somatic if it satisfies the following criterion:

$$c_E \leq n_A \leq c_H,$$

where $n_A$ is the number of non-reference allele reads at the locus, and $c_E$ and $c_H$ are integers which maximise:

$$p_n = r_{RR} \sum_{n_A = c_E}^{c_H} \frac{n}{n_A} p_A^{n_A} (1 - p_A)^{n - n_A}$$

subject to:

$$\frac{r_c}{1 - r_c} \frac{p_n}{r_{RR}\alpha_E + r_H\alpha_H} \geq g_r$$

with $r_{RR} = \frac{1}{2}(1 - r_H + \sqrt{1 - 2r_H})$, and $p_A = \left(1 - \frac{4}{3}\varepsilon\right)f + \varepsilon$. Here $n$ is the sum of reference and alternate allele depths at the locus, $r_H$ is the expected rate of heterozygous variant germline loci, $r_C$ the expected rate of somatic variant loci, $\varepsilon$ the base read error rate, $g_r$ the minimum acceptable ratio of true positive calls to false positive, and $f$ the expected somatic variant allele fraction. $\alpha_E$ and $\alpha_H$ are test sizes corresponding to thresholds $c_E$ and $c_H$: $c_E \equiv \inf\{n_A : \Pr(\text{err}) < \alpha_E\}$, $c_H \equiv \sup\{n_A : \Pr(\text{het}) < \alpha_H\}$, $\Pr(\text{err}) = \sum_{i=n_A}^{n} \frac{n}{i} \varepsilon^i (1 - \varepsilon)^{n-i}$, $\Pr(\text{het}) = \sum_{i=0}^{n_A} \frac{n}{i} \left(\frac{1}{2} + \frac{1}{3}\varepsilon\right)^i \left(\frac{1}{2} - \frac{1}{3}\varepsilon\right)^{n-i}$. Informally, this procedure selects variants with an alternate allele count $n_A$ too large to be due to sequencing error ($n_A \geq c_E$), yet too low to be from a poorly sampled heterozygous germline locus ($n_A \leq c_H$). The derivation of this procedure and further details are available in Supplementary Data 5. A notable advantage of this procedure is that it yields per-locus estimates of variant detection sensitivity, as $p_n$. These estimates are critical in normalisation of variant detection rates to account for differential coverage across variable sequencing runs, which is necessary for the accurate estimation of sample somatic variant burden.

Insufficiencies of the simple error model used above result in incomplete control of the signal to noise ratio, and further filtering is required to reliably quantify somatic variant burden. Assuming that true somatic events and false-positive machine noise exhibit differential sequence context bias, we use a spectral dimensionality reduction approach to achieve additional denoising and summarise the total somatic variant burden in each sample. Extending previous cancer somatic signature work[66,67], we calculate per-sample sensitivity-normalised somatic variant burden for each of 96 single-nucleotide variant classes, as the total number of detected somatic events of a given class in that sample, divided by the sum of $p_n$ in that sample for all loci corresponding to the given variant class. The resulting $96 \times n$ normalised burden matrix is then reduced by non-negative matrix factorisation, using 100 random optimisation starting points per cardinality. To select the appropriate factorisation cardinality, we reduce the burden matrix by merging groups of 16 age-consecutive samples by summing burden for each variant class, and factorise this reduced matrix with 100 random restarts and cardinality ranging from 2 to 10. The lowest cardinality that gives an inflection point on the plot of explained variance vs cardinality is selected, and applied to the full burden matrix. Per-sample scores are extracted from the best of 100 random runs at this final selected cardinality.

We applied the above procedure to the MGRB and ASRB samples, with parameters adapted to maximise sensitivity with low-depth sequencing data: $g_r = 5$, $f = 0.2$, $\varepsilon = 2.0 \times 10^{-3}$, $r_H = 1.0 \times 10^{-4}$, $r_C = 5.0 \times 10^{-7}$. Our filtering process employed SNVs identified by samtools mpileup, with maximum depth 101, mapping quality adjustment of 50, BAQ recalculation, no indel reporting, and minimum read and mapping qualities of 30, and employed a blacklist of common SNPs observed in either MGRB or dbSNP. The factorisation cardinality procedure applied to our data indicated that three signatures best described the mutation patterns observed (Supplementary Fig. 11). Signature 3 in this work was quantitatively similar to COSMIC signature 5 (cosine similarity 0.81), previously reported to be associated with age at cancer diagnosis[29], and the per-sample scores for this signature were used as the summative somatic burden measure. Signature 1 from this work was very similar to COSMIC signature 1 (cosine similarity 0.95), which has also been associated with spontaneous deamination processes and age. However, we observed substantial inter-cohort differences in score distribution for this signature, suggestive of high technical variability, and did not examine it further.

**Somatic copy number variants**. We developed a model-based strategy to identify subclonal copy number variants (CNVs), assuming a single genetically homogeneous subclone present on a background of diploid cells.

We first defined a set of autosomal SNPs with highly stable sequencing characteristics on our platform. We selected loci containing autosomal biallelic SNPs in the MGRB cohort, with a variant allele fraction between 5% and 95%, and a mean GC content in the surrounding 100 bp of between 30% and 55%. These were further filtered to retain only loci with highly consistent coverage in both the MGRB and ASRB cohort data, with mean($DP_{rel}$) $\in [0.9, 1.1]$ in both cohorts, var($DP_{rel}$) $\in [0.025, 0.033]$ in the MGRB, and var($DP_{rel}$) $\in [0.025, 0.040]$ in the ASRB cohort. Here $DP_{rel}$ is locus depth relative to mean sample depth, and statistics are calculated over all samples in each cohort. In total, 1,862,065 loci passed all filters, with a median inter-locus distance of 626 bp, and 5th and 95th percentiles of 30 and 4904 bp, respectively.

We individually genotyped MGRB, 45 and Up cancer, and ASRB samples at this set of highly reliable loci using GATK HaplotypeCaller with default parameters, except for a variant window size of 100 bp (-ip 100). Within each sample, the depths of reference and variant alleles at all heterozygous SNV target loci were fit to the following subclonal CNV model to produce estimates of local ploidy and global sample subclonal fraction.

Consider a locus $i$ in a single sample which contains fraction $f$ of aneuploid cells, the remaining $1 - f$ being entirely diploid (gonosomes are not modelled). We denote the copy number (ploidy) of the aneuploid cells at $i$ with $k_{1,i}$ and $k_{2,i}$, $k_{.,i} \in \aleph$. For example, $k_{1,i}, k_{2,i} = (1, 1)$ denotes a diploid state (no aneuploidy), $k_{1,i}, k_{2,i} = (1, 0)$ the deletion of one allele, and $k_{1,i}, k_{2,i} = (2,2)$ duplication of both alleles. Our task is to estimate $k_{1,i}, k_{2,i}$ for all $i$, and $f$ globally, given reference and non-reference allele depths $d_{R,i}$ and $d_{A,i}$.

The extent to which the aneuploid cell ploidies $k_{1,i}$ and $k_{2,i}$ affect the representation of alleles in the mixed cell population depends on the aneuploid cell fraction $f$. Let $p_{1,i}$ and $p_{2,i}$ represent the mean ploidy of each chromatid in the mixed cell DNA pool. As the pool is assumed to consist of only two populations, with $1 - f$ of the cells diploid, $p_{1,i} = fk_{1,i} + (1 - f)$ and $p_{2,i} = fk_{2,i} + (1 - f)$.

We assume that the sequencer does not exhibit allelic bias. Then, $E[d_{1,i}] = c_i p_{1,i}$, $E[d_{2,i}] = c_i p_{2,i}$, with $c_i$ a normalising constant to account for the depth at locus $i$. Here $d_{1,i}$ and $d_{2,i}$ denote the depths of reads from chromatid 1 and 2, respectively. Unfortunately we do not have phased genotypes, and so cannot easily determine the chromatid source of each read. Instead we have unphased reference and non-reference depths $d_{R,i}$ and $d_{A,i}$, and must account for the resulting phase uncertainty with a mixture model.

Disregarding allele phasing we model the depths of reference and non-reference reads at $i$ using a mixture: $d_{R,i}, d_{A,i} \sim D(c_i p_{1,i}), D(c_i p_{2,i})$ with probability $\frac{1}{2}$, else $d_{R,i}, d_{A,i} \sim D(c_i p_{2,i}), D(c_i p_{1,i})$, $D(\mu)$ denoting a distribution function with expected value $\mu$. In our implementation we employ a negative binomial distribution for $D$, with probability mass function $f_D(x; \mu, s) \equiv \frac{\Gamma(x+s)}{\Gamma(x+1)\Gamma(s)} q^s (1 - q)^x$, $q \equiv \frac{s}{s+\mu}$. The size term $s$ captures overdispersion relative to the Poisson distribution, and is optimised per-sample in the model fit.

The normalising constant $c_i$ is half the expected depth at locus $i$, which is itself a complex function of locus- and sample-specific factors. We model this function at the locus- and sample-level using empirical cohort depth measurements, and a sample-specific GC bias correction. Specifically, we define $c_i \equiv b_i e^{h(g_i)}$, where $b_i$ is the mean relative depth of locus $i$ (where relative depth is defined as $d_i \div \frac{1}{n}\sum_i d_i$, with $d_i \equiv d_{R,i} + d_{A,i}$ and $n$ the number of target loci, $n = 1,862,065$), across the sample's cohort (either MGRB or ASRB), $h$ is a smooth function, and $g_i$ is a vector of GC fraction in windows of various size around locus $i$. For this work, $g_i$ was a 5-vector of GC fraction in windows of size 100, 200, 400, 600, and 800 bp, calculated on the reference sequence centred at locus $i$. The sample-specific GC correction function $h$ was implemented using a generalised additive model (GAM) with five smooth terms, and fit to all heterozygous loci for each sample as $\ln(c_i \div b_i) \sim s(r_{1,i}) + s(r_{2,i}) + s(r_{3,i}) + s(r_{4,i}) + s(r_{5,i})$, with $r_{j,i}$ being the score of the $j$th principal component of the GC fraction matrix for locus $i$, and $s$ denoting a penalised regression spline term. GAMs were fit using mgcv 1.8–17 (ref. [68]) with default parameters.

A greedy agglomerative algorithm was used to segment the genome of each sample into regions of differing ploidy state. Initially the genome was divided into segments of 100 consecutive heterozygous loci, with segment boundaries enforced between chromosomes. Adjacent segments were tested for identical distribution of $d_{R,i} \div c_i$, $d_{A,i} \div c_i$ by a two-sample Kolmogorov–Smirnov test, and the two segments with the highest $p$ value genome-wide were merged. This process was repeated until either no segment pairs remained to merge, or all Kolmogorov–Smirnov test $p$ values were less than 0.01. Segments were never merged between chromosomes.

The above model was fit to the allele counts within each genome segment by maximum likelihood. Ploidies of each segment, $k_{1,i}$ and $k_{2,i}$, as well as the global aneuploid fraction $f$ and overdispersion $s$, were optimised by grid search with local polishing. As very high ploidies coupled with low $f$ result in highly expressive but likely incorrect models, the maximum allowable ploidy $k_{max}$; $k_{1,i}, k_{2,i} \leq k_{max}$ was determined in an outer loop through minimisation of the Bayesian Information Criterion (BIC). A final polishing step was applied to the BIC-optimal model, which merged consecutive segments of the genome if they were assigned identical chromatid ploidies by the model. This final polished model yielded global cell fraction $f$, as well as local ploidies across the genome, for the single aneuploid clone assumed to be present in each sample.

**Clonal haematopoiesis**. Extending previous work[6], clonal haematopoiesis of indeterminate potential (CHIP) was defined in an individual if either: a somatic small variant (see section Incidental somatic variant detection) was detected with a variant allele frequency of at least 10%, or somatic CNV (see section Somatic copy number variants) indicated the presence of a clone comprising at least 10% of nucleated blood cells.

**Somatic burden statistical analysis**. Exploratory analysis indicated that variable transformation was required for some measures. For the following analyses, telomere length and Y copy number were modelled as-is; somatic variant burden, mitochondrial load, and mitochondrial variant count were log-transformed prior to modelling; and grip strength in kg was power transformed with exponent 0.7.

Within-cohort trends in somatic measures were estimated by linear regression, with 95% Wald confidence intervals. Likelihood ratio tests of nested models were used to evaluate inter-cohort trend differences, with $p$ values corrected for multiple testing by Holm's step-up procedure[69].

We used a permutation procedure to test the importance of somatic burden measures in predicting grip strength and gait speed, conditioned on age. For each of 18 possible frailty measure × somatic measure × sex combinations (frailty measures: grip strength, gait speed; Somatic measures: Telseq telomere length, nuclear somatic variant burden, mtDNA copy number, mitochondrial variant count, and Y copy number in males only), we calculated the deviance of the following generalised additive model frailty $\sim s(\text{age}) + s(\text{weight}) + s(\text{BMI}) + s(\text{abdocirc}) + s(\text{somatic})$, with age in years, weight in kg, BMI in $kg/m^2$, abdominal circumference (abdocirc) in cm, and the somatic measure of interest (transformed if relevant following exploratory analysis). In this model specification, $s(x)$ denotes a GCV-penalised thin plate spline smooth term in $x$ as implemented in R package mgcv 1.8-17(ref. [68]), with Gaussian error and identity link. This model's deviance $d$ was compared to the deviance $d_{(i)}$ of 10,000 models fit in the same manner but with the somatic variable permuted, and a $p$ value estimated as $\hat{p} = \frac{1}{10,001} \left( \sum_i \left[ d_{(i)} \leq d \right] + 0.5 \right)$. To address multiple testing concerns we used a two-stage process. In the first stage $p$ values were calculated as above for all 18 tests on a randomly selected subset of 25% of the ASPREE samples. Tests with a $p$ value less than 0.2 in the first stage were tested in the second validation stage on the remaining 75% of the ASPREE samples, and these second-stage $p$ values corrected for multiple testing by Holm's method.

We observed cohort differences in intercepts in plots of somatic measures vs age. To remove these solely for the purposes of illustration (Fig. 3), for each somatic measure we fit the generalised additive model measure $\sim s(\text{age, by} = \text{sex}) + \text{cohort}$, with Gaussian error and identity link. In this model specification s(age, by = sex) denotes a GCV-penalised thin plate spline with age as the predictor variable, stratified by sex. Model fits were performed using the R package mgcv[68]. After confirming the suitability of the model fits, cohort-specific effects were removed by calculating the quantity $y_i' = y_i - \hat{s}_C + \hat{s}_{ASPREE}$ for each individual and measure, where $y_i'$ is the cohort-corrected somatic measure for individual $i$, to be plotted; $y_i$ is the original measurement for individual $i$ in cohort $C$; and $\hat{s}_C$ and $\hat{s}_{ASPREE}$ are the model estimates of the cohort intercept term for cohort $C$ and the ASPREE cohort, respectively. In this manner, somatic measurements were transformed to have an intercept matching that fitted to the ASPREE cohort.

We used the following procedure to illustrate the effect of mtDNA copy number on grip strength in males. For each male individual $i$ in the ASPREE cohort, an age-local quantile of mitochondrial DNA copy number $c_i$ was defined as $q_i \equiv \hat{F}_i(c_i)$, where $\hat{F}_i$ is the empirical cumulative distribution function of $c$ in the neighbourhood of individual $i$, with the neighbourhood of an individual $i$ defined as all male ASPREE individuals within ±1 year of age of $i$. Ages were rounded to the nearest integer for the purposes of neighbourhood definition; for the median ASPREE male age of 80 years, this neighbourhood contained 293 men with ages in [79, 81] years. Given these local mtDNA copy number quantile estimates $q$, a generalised additive model of the form grip strength$\sim$age $+ s(q)$ was fit using the R package mgcv[68], with

$s$ smooth term as above. Predictions from this model with age = 80 and varying $q$ defined the estimated influence of age-local mtDNA copy number on grip strength for an 80 year old man. These grip strength predictions were transformed to effective age estimates assuming typical mtDNA copy number by inversion of the model predictions for $s = 0.5$, and used to calculate an age excess as a function of $q$. Variability of this relationship was estimated using 100,000 bootstrap samples, and results presented as highest posterior density intervals.

**Reporting summary**. Further information on research design is available in the Nature Research Reporting Summary linked to this article.

## Data availability

Summary variant frequency data for the MGRB cohort are available at the web portal: https://sgc.garvan.org.au/explore. Raw genomic data have been deposited at the European Genome-Phenome Archive under study ID EGAS00001003511. Phenotype data are available upon application to the MGRB Data Access Committee at mgrb@garvan.org.au.

## Code availability

Source code for all analyses is available at https://github.com/mpinese/mgrb-manuscript; source code for the somatic SNV and LoH detection tools can be found at https://github.com/mpinese/soma-snv and https://github.com/mpinese/soma-cnv.

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

## Acknowledgements

Whole-genome sequencing of the MGRB, ASRB, and 45 and Up cancer cohort was undertaken at the Kinghorn Centre for Clinical Genomics, which is supported by the Kinghorn Foundation. Sequencing of these cohorts was funded through the NSW Genomics Collaborative Grants scheme from the NSW Office for Health and Medical Research. Processing and warehousing of genomic data employed resources and services from the National Computational Infrastructure (NCI), which is supported by the Australian Government. The authors acknowledge the ASPREE Healthy Ageing Biobank, ASPREE Investigator Group, and ASPREE Collaborating Practitioners listed on www.aspree.org. ASPREE was funded by the National Institute on Aging and the National Cancer Institute at the National Institutes of Health (grant number U01AG029824); the National Health and Medical Research Council of Australia (grant numbers 334047, 1127060); the Victorian Cancer Agency and Monash University (Australia). The ASPREE Healthy Ageing Biobank was supported by the Commonwealth Scientific and Industrial Research Organisation (Australia), the Victorian Cancer Agency (Australia), and Monash University (Australia). We acknowledge the dedicated and skilled staff in Australia and the U.S. for the conduct of the ASPREE trial, and the ASPREE participants who willingly volunteered. This research was completed using data collected through the 45 and Up Study (www.saxinstitute.org.au). The 45 and Up Study is managed by the Sax Institute in collaboration with major partner Cancer Council NSW; and partners: the National Heart Foundation of Australia (NSW Division); NSW Ministry of Health; NSW Government Family & Community Services—Ageing, Carers and the Disability Council NSW; and the Australian Red Cross Blood Service. We thank the many thousands of people participating in the 45 and Up Study. Genomic analysis of the Australian Schizophrenia Research Bank (ASRB) was supported by a New South Wales Health, Collaborative Genomics grant programe (to M.J.C., M.J.G., V.J.C.), a NARSAD Independent Investigator Grant (to M.J.C.) and National Health and Medical Research Council

(NHMRC) project grants (1067137, 1147644, 1051672). Samples were also collected by the ASRB with the support of the NHMRC, the Pratt Foundation, Ramsay Health Care, and the Viertel Charitable Foundation. The ASRB was supported by the Schizophrenia Research Institute (Australia), utilising infrastructure funding from NSW Health and the Macquarie Group Foundation. This research has been conducted using the UK Biobank Resource under Application Number 17984. D.M.T is the recipient of an NHMRC Principal Research Fellowship (RegKey:1104364). M.J. Cairns was supported by an NHMRC Senior Research Fellowship (#1121474). M.J. Cowley was supported by a NSW Health Early-Mid Career Fellowship. J.R.A. was supported by University of Newcastle RHD and an Emlyn and Jennie Thomas Postgraduate Medical Research Scholarship. M. E.D., C.P., W.K., D.D., and S.H. were supported by the Kinghorn Foundation. E.M.R was supported by the Vodafone Foundation. The authors thank Prof. Christopher Goodnow, Prof. Diane Fatkin, Dr. Catherine Vacher for helpful comments and discussion, Dr. Eleni Giannoulatou for the atrial fibrillation polygenic score coefficients, and Verity Hodgkinson for biospecimen management.

## Author contributions

Conceptualisation: M.P., M.E.D., and D.M.T.; methodology: M.P., E.M.R., J.E.P., C.P., G. G., and D.M.T.; software: M.P., E.M.R, and A.L.S; formal analysis: M.P., E.M.R., A.L.S., M. R., C.P., M.J. Cowley, and P.A.J.; investigation: V.F.S.K. and H.A.P.; resources: P.L., A.A., M.B., M.McN., E.B., M.R.N., C.M.R., A.M.M., R.C.S., R.W., M.J. Cairns, H.M.C., J.R.A., C. F., M.J.G., V.J.C., S.N., G.G., R.L.W., U.G., S.R., and J.J.McN.; data curation: M.P., P.L., M. B., M. McN., and T.J.N.C.; writing—original draft, M.P.; writing—review & editing: M.P., P.L., and D.M.T.; visualisation: S.H., D.D., and W.K.; project administration: A.S. and M-J. B.; funding acquisition: M.E.D. and D.M.T.; supervision: M.E.D. and D.M.T.

## Competing interests

The authors declare no competing interests.

## Additional information

Mark Pinese[1,2,3,26], Paul Lacaze[4,26], Emma M. Rath[1], Andrew Stone[1,2,3,5], Marie-Jo Brion[1], Adam Ameur[4,6], Sini Nagpal[7], Clare Puttick[1], Shane Husson[1], Dmitry Degrave[1], Tina Navin Cristina[8], Vivian F.S. Kahl[9], Aaron L. Statham[1], Robyn L. Woods[4], John J. McNeil[4], Moeen Riaz[4], Margo Barr[10], Mark R. Nelson[4,11], Christopher M. Reid[4,12], Anne M. Murray[13,14], Raj C. Shah[15], Rory Wolfe[4], Joshua R. Atkins[16,17], Chantel Fitzsimmons[16,17], Heath M. Cairns[16,17], Melissa J. Green[18,19], Vaughan J. Carr[18,19,20], Mark J. Cowley[1,2,3], Hilda A. Pickett[9], Paul A. James[21,22], Joseph E. Powell[23,24], Warren Kaplan[1,5], Greg Gibson[7], Ulf Gyllensten[6], Murray J. Cairns[16,17], Martin McNamara[8], Marcel E. Dinger[1,25,26] & David M. Thomas[1,5,26]*

[1]Garvan Institute of Medical Research, Sydney, NSW, Australia. [2]Children's Cancer Institute, University of New South Wales, Sydney, NSW, Australia. [3]School of Women's and Children's Health, Faculty of Medicine, University of New South Wales, Sydney, NSW, Australia. [4]Department of Epidemiology and Preventive Medicine, Monash University, Melbourne, VIC, Australia. [5]St Vincent's Clinical School, Faculty of Medicine, University of New South Wales, Sydney, NSW, Australia. [6]Science for Life Laboratory, Department of Immunology, Genetics and Pathology, Uppsala University, Uppsala, Sweden. [7]Center for Integrative Genomics, Georgia Institute of Technology, Atlanta, GA, USA. [8]Sax Institute, Sydney, NSW, Australia. [9]Children's Medical Research Institute, Faculty of Medicine and Health, University of Sydney, Westmead, NSW, Australia. [10]Centre for Primary Health Care and Equity, University of New South Wales, Sydney, NSW, Australia. [11]Menzies Institute for Medical Research, University of Tasmania, Hobart, TAS, Australia. [12]School of Public Health, Curtin University, Perth, WA, Australia. [13]Berman Center for Outcomes and Clinical Research, Hennepin Healthcare Research Institute, Hennepin Healthcare, Minneapolis, MN, USA. [14]Division of Geriatrics, Department of Medicine, Hennepin County Medical Center and University of Minnesota, Minneapolis, MN, USA. [15]Department of Family Medicine and Rush Alzheimer's Disease Center, Rush University Medical Center, Chicago, IL, USA. [16]School of Biomedical Sciences and Pharmacy, The University of Newcastle, Callaghan, NSW, Australia. [17]Centre for Brain and Mental Health Research, Hunter Medical Research Institute, Newcastle, NSW, Australia. [18]School of Psychiatry, University of New South Wales, Sydney, NSW, Australia. [19]Neuroscience Research Australia, Sydney, NSW, Australia. [20]Department of Psychiatry, School of Clinical Sciences, Monash University, Melbourne, VIC, Australia. [21]Parkville Familial Cancer Centre, Peter MacCallum Cancer Centre, Melbourne, VIC, Australia. [22]Sir Peter MacCallum Department of Oncology, University of Melbourne, Melbourne, VIC, Australia. [23]UNSW Cellular Genomics Futures Institute, School of Medical Sciences, University of New South Wales, Sydney, NSW, Australia. [24]Garvan-Weizmann Centre for Cellular Genomics, Garvan Institute of Medical Research, Sydney, NSW, Australia. [25]School of Biotechnology and Biomolecular Sciences, Faculty of Science, University of New South Wales, Sydney, NSW, Australia. [26]These authors contributed equally: Mark Pinese, Paul Lacaze, Marcel E. Dinger, David M. Thomas. *email: d.thomas@garvan.org.au

