## [Peer Review File · Nature Communications]

Reviewers' Comments:

Reviewer #1:

Remarks to the Author:

The manuscript by Pinese et al. reports the results from a whole-genome sequencing (WGS) study of healthy individuals above 70 years of age. The authors performed several analyses that show that the created dataset can be considered a healthy control dataset. Given that the manuscript presents a unique dataset that will be of use as reference dataset for many future genetic studies (with a very nicely designed web portal), it warrants rapid publication in Nature Communications.

I only have some relatively minor comments:

- In general, a polygenic score (PS) is created based on genome-wide significant genetic variants associated with a specific phenotype. However, at least for the "short lifespan" phenotype it seems that the authors used results of a subset of genetic variants that showed a suggestive association ($P \leq 1 \times 10^{-5}$) with longevity (i.e. survival to ages above 90 years), of which some could not be replicated in the replication phase of the same paper. Hence, I think the authors should either remove this PS from their manuscript or create a new PS that is based on the variants that remained suggestive significant in the joint analysis of the discovery and replication phase (i.e. rs2149954, rs2784505, and rs4420638). Good alternative PSs for "short lifespan" would be ones that are created based on the GWA study results from Timmers (<https://elifesciences.org/articles/39856>) or Zenin and colleagues (<https://www.nature.com/articles/s42003-019-0290-0>), given that they identified multiple genetic variants associated with parental lifespan and healthspan at a genome-wide significant level. The authors should also revise the part on the PS estimation in their Methods section and describe how the selection of variants for the created PSs was performed (i.e. are they based on genome-wide significant genetic variants or have some of them be created using other P-value selection criteria).
- I do not see the added value of testing the correlation of the somatic burden measures with grip strength and gait speed, it does not add so much to the manuscript. I do not see how these results help to "derive quantitative measures of biological ageing from standard-depth WGS" (lines 432-433), given that they only observed a very minor correlation between the count of mitochondria and grip strength in men, which they could not really explain. In my view this whole part could be removed from the manuscript to make the message more straightforward.
- Lines 82-83; The statement "consistently associated the APOE and FOXO3A loci with lifespan and longevity" is not correct. The only locus that has consistently been associated at a genome-wide significant level is APOE. FOXO3 has not (yet) been reported to meet this criterion in GWAS. Hence, the authors should update this statement and maybe refer to a more recent review on the genetics of longevity and lifespan, since the review from Brooks-Wilson is outdated.
- Lines 279-280; The authors mention that "low depth WGS has not been applied to the detection of CHIP". However, this is not true. We have previously used WGS in long-lived individuals to detect CHIP (van den Akker et al. Blood 2016), so the authors should change this statement accordingly.
- It would be nice if the authors can provide a Figure in which they show the number of clonal somatic variants they have detected in genes implicated in hematologic cancers (similar to Jaiswal et al. NEJM 2014, Figure 2A).
- The numbering of the Supplementary Figures should be updated, currently Supplementary Figures 5 and 6 are mentioned before Supplementary Figure 4.
- Supplementary Figure 1; There is an error in the legend of this figure, i.e. "Eurpoean" should be

“European”.

- Supplementary Figure 3; “Strand-specific...only shown” should be changed into “Only strand-specific...are shown”.

Joris Deelen

Reviewer #2:

Remarks to the Author:

The authors present a comprehensive whole genome analysis of healthy aging individuals. A number of interesting observations are presented, with some novelty on the somatic mutation analysis, however much of the other findings are anticipated by prior studies and/or questionable due to the lack of a well matched control. Regardless, the dataset is a useful resource for future studies. Some specific comments below:

The authors emphasize in their comparison to Erikson et al. 2016 that no depletion of rare pathogenic variants was observed previously and are observed now. As far as I can tell, this does not appear to be true. A depletion of rare pathogenic variants is observed relative to disease cases - but no depletion is observed relative to a population control. A depletion relative to disease cases is not a compelling argument for this dataset as a super control. A depletion relative to a population control would be required, and this is not observed just as it is not observed in Erikson et al.

An enhancement relative to Erikson et al. is the inclusion of structural variants. However, there is no easy way to evaluate the structural variant findings. Are they more or less than expected by chance or in-line with what is expected in the general population. The issue here is there is no matched control population with raw data processed in the same way as the MGRB cohort.

Similarly, the polygenic comparisons are concerning because of the difference in background population. While a reduction in polygenic risk is expected, its not clear what the influence of Australian vs UK vs a more heterogeneous gnomAD would be. It is known that there is population substructure that can lead to differences in polygenic risk, even among European individuals. An appropriate control would likely be Australian, not European.

The somatic and mitochondrial analyses are quite interesting and novel. However, these processes, as stated by the authors, are heavily age-related. Without a age-matched control dataset, it is difficult to know whether the changes in any of these parameters are different in the well elderly vs normal elderly. The association with grip strength internal to the MGRB is interesting.

Overall, the manuscript represents a useful dataset, with some concerns regarding the novelty and validity of findings.

Reviewer #3:

Remarks to the Author:

This is a timely and interesting paper by Pinese et al. profiling the genetic characteristics of a cohort of “healthy elderly” who reached the age of 70 free of cancer, cardiovascular disease, and dementia. Timely because there are a number of papers now available investigating lifespan and healthspan, but these focus on common variation using genotyping arrays, and thus not only cannot analyse rare or structural variation in detail, but the approach taken by Pinese et al. also means that somatic variation can be analysed. This is the real novelty of this paper, in my opinion,

and is deserved of publication. My comments and suggestions are included below. Note: I have not attempted to review the methods for the sequencing analysis, as this is beyond my area of expertise.

1. (Introduction, lines 81-84) Although the APOE locus has been consistently associated with lifespan and longevity, the FOXO3A locus is much less consistent. Both the Broer and Deelen 2014 GWAS (PMIDs 25199915 and 24688116) did not find strong evidence when comparing older individuals to younger controls (although the Broer paper claims it, the FOXO3A evidence is weak by comparison). Neither did Pilling or Timmers (2017, 2019) when looking at parents survival using UK Biobank and the LifeGen consortium, respectively (PMIDs 29227965 and 30642433). I suggest revising this section to reduce the emphasis on FOXO3A, and mention the loci arising from the Timmers paper especially (includes CDKN2A/B locus, HLA, etc.).

2. (Results, lines 161-164) It is interesting that there are individuals in the cohort carrying potentially pathogenic variants. Is this prevalence comparable to or lower than other general studies (i.e. are they less common in healthy elderly)? I realise there is an analysis of pathogenic variants overall in the next section (starting line 188) but this does not seem to be the same question.

3. (Results, lines 188-207) Could this analysis be repeated using the genes associated with lifespan in the Timmer 2019 analysis (PMID 30642433)? I.e. they identified common variants associated with (parents) survival near ~12 genes, it would be interesting to know whether these genes carry fewer rarer/pathogenic variants in the healthy elderly than population controls. The authors could consider the same analysis for the loci identified by Zenin et al. in their GWAS of healthspan (PMID 30729179). Similar to the cancer/CAD loci already investigated, these loci have strong a priori evidence for involvement in healthy ageing.

4. Figure 1a. I think this would be improved by including the CAD results, even though they are null, for completeness (would be good to see estimate and CIs in the text at the very least). This would also help the general reader, to emphasise that it appears that rare variants for cancer are depleted, but no evidence was found (here) for CAD variants.

5. (Results, lines 227-230) Although it is stated in the methods, it might be useful to here recap what specific criteria were made for the MGRB. Is it here just referring to individuals with no cancer, cardiovascular disease, and dementia, or are the other criteria from the ASPREE study also referred to (which includes "no anaemia", etc). To the casual reader it will be assumed to just be the former criteria, given the abstract.

6. Statements about associations (significant or not) should include the relevant test statistic. For example Results section on polygenic scores. I realise some of this information is displayed graphically on Figure 1b but it should be available in the text, with additional information in the supplement. (Incidentally there is a typo on line 237, it should be Figure 1b not 2b). For example line 239 states the score for short lifespan (from the Deelen paper) "was significantly depleted in MGRB relative to gnomAD and UKB." The authors should report by how much (and how significant) this difference was, otherwise it is impossible to determine whether this is meaningful. This may be better expressed the other way: "The MGRB healthy elderly harboured XX more lifespan-increasing alleles compared to gnomAD and UKB populations (beta, CIs, p...)." The figure contains information on a Normalised score (AU) which is useful to compare the different scores (i.e. to see that Atrial Fibrillation has a larger effect than breast cancer – if I am interpreting it correctly) but is not necessarily that interpretable.

7. The section on somatic mutation accumulation with age is very interesting. Table 3, comparing the accumulation of different measures is the highlight of this analysis, accompanied with the analysis on functional outcomes (i.e. grip strength and gait). Given that the WGS is of blood, were any immune-relevant parameters available e.g. c-reactive protein, infections, lymphomas? The

effect of mtDNA copy number on grip in men is very weak, so a more "relevant" outcome may be better powered. This section is again lacking detail however: lines 338 to 341 imply that all the somatic change measures were tested against grip and gait, yet these results are not presented. These should be included as a supplementary table.

8. Related to point 7. I think the "Summary" (lines 60-61) should better reflect the actual findings. Unless I have misinterpreted the text, it is misleading to state that "Pervasive age-related somatic changes were correlated with grip strength in men" when the p-values for mtDNA copy number~grip are very nominal (0.051 in 1st stage, 0.036 in 2nd) when clearly many statistical tests have been performed (5x somatic change measures, 2x outcomes, stratified by sex). These statements (and the accompanying conclusions/discussion) should be appropriately moderated.

Minor comments

1. It would be useful to include a title and some accompanying information at the start of each supplementary file, especially the tables. Currently it is unclear at points which table is which, and what the columns refer to.
2. "UK BioBank" just has one capitalized "B" i.e. should be "UK Biobank" (see www.ukbiobank.ac.uk)

Reviewer 1

The manuscript by Pinese et al. reports the results from a whole-genome sequencing (WGS) study of healthy individuals above 70 years of age. The authors performed several analyses that show that the created dataset can be considered a healthy control dataset. Given that the manuscript presents a unique dataset that will be of use as reference dataset for many future genetic studies (with a very nicely designed web portal), it warrants rapid publication in Nature Communications.

I only have some relatively minor comments:

1.1 - In general, a polygenic score (PS) is created based on genome-wide significant genetic variants associated with a specific phenotype. However, at least for the "short lifespan" phenotype it seems that the authors used results of a subset of genetic variants that showed a suggestive association ($P \leq 1 \times 10^{-5}$) with longevity (i.e. survival to ages above 90 years), of which some could not be replicated in the replication phase of the same paper. Hence, I think the authors should either remove this PS from their manuscript or create a new PS that is based on the variants that remained suggestive significant in the joint analysis of the discovery and replication phase (i.e. rs2149954, rs2784505, and rs4420638). Good alternative PSs for "short lifespan" would be ones that are created based on the GWA study results from Timmers (<https://elifesciences.org/articles/39856>) or Zenin and colleagues (<https://www.nature.com/articles/s42003-019-0290-0>), given that they identified multiple genetic variants associated with parental lifespan and healthspan at a genome-wide significant level. The authors should also revise the part on the PS estimation in their Methods section and describe how the selection of variants for the created PSs was performed (i.e. are they based on genome-wide significant genetic variants or have some of them be created using other P-value selection criteria).

Thank you for bringing to our attention the Timmers and Zenin papers, which were published after our original submission. In Figure 1 we have replaced the original Deelen 2014 scores with these newer scores. We have also clarified in the methods exactly how our scores were derived from published summary statistics. In this section we have emphasised that our polygenic scores were largely based on genome-wide significant loci. Although this will probably lead to suboptimal scores, it was a practical choice given that many of the source publications did not publish their full genome-wide summary statistics.

1.2 - I do not see the added value of testing the correlation of the somatic burden measures with grip strength and gait speed, it does not add so much to the manuscript. I do not see how these results help to "derive quantitative measures of biological ageing from standard-depth WGS" (lines 432-433), given that they only observed a very minor correlation between the count of mitochondria and grip strength in men, which they could not really explain. In my view this whole part could be removed from the manuscript to make the message more straightforward.

We agree that our observations of an age-conditioned association between somatic and functional measures are subtle. Given the diverse opinion on this section, the complex nature of the analysis, and the subtle effect, we have offered to the Editor to remove this particular element from the manuscript if he or she believes it is necessary.

1.3 The statement “consistently associated the APOE and FOXO3A loci with lifespan and longevity” is not correct. The only locus that has consistently been associated at a genome-wide significant level is APOE. FOXO3 has not (yet) been reported to meet this criterion in GWAS. Hence, the authors should update this statement and maybe refer to a more recent review on the genetics of longevity and lifespan, since the review from Brooks-Wilson is outdated.

Thank you for correcting our misconception on this point. We have updated the manuscript and cited a more recent review on the subject.

1.4 The authors mention that “low depth WGS has not been applied to the detection of CHIP”. However, this is not true. We have previously used WGS in long-lived individuals to detect CHIP (van den Akker et al. Blood 2016), so the authors should change this statement accordingly.

We apologise for our omission of this work. We have cited the aforementioned study, and modified the manuscript to focus on our detection of somatic copy number alterations by WGS, which we believe to be a novel aspect.

1.5 - It would be nice if the authors can provide a Figure in which they show the number of clonal somatic variants they have detected in genes implicated in hematologic cancers (similar to Jaiswal et al. NEJM 2014, Figure 2A).

This figure has been added, as Supplementary Figure 6.

1.6 - The numbering of the Supplementary Figures should be updated, currently Supplementary Figures 5 and 6 are mentioned before Supplementary Figure 4.

- Supplementary Figure 1; There is an error in the legend of this figure, i.e. “Eurpoean” should be “European”.

- Supplementary Figure 3; “Strand-specific...only shown” should be changed into “Only strand-specific...are shown”.

Thank you for your close reading of the manuscript; these errors have been corrected.

Reviewer 2

The authors present a comprehensive whole genome analysis of healthy aging individuals. A number of interesting observations are presented, with some novelty on the somatic mutation analysis, however much of the other findings are anticipated by prior studies and/or questionable due to the lack of a well matched control. Regardless, the dataset is a useful resource for future studies. Some specific comments below:

2.1 The authors emphasize in their comparison to Erikson et al. 2016 that no depletion of rare pathogenic variants was observed previously and are observed now. As far as I can tell, this does not appear to be true. A depletion of rare pathogenic variants is observed relative to disease cases - but no depletion is observed relative to a population control. A depletion relative to disease cases is not a compelling argument for this dataset as a super control. A depletion relative to a population control would be required, and this is not observed just as it is not observed in Erikson et al.

We believe the reviewer is referring to the second paragraph of the discussion, in which we state “the MGRB reveals a striking depletion in disease-associated common and rare variation, relative to both affected cases, as well as datasets frequently used as controls in genetic studies.” Upon re-reading this section we acknowledge that it implies that both rare and common variation were depleted in MGRB relative to gnomAD, and this is indeed not the case. We have accordingly reworked this paragraph to make the distinction clear.

2.2 An enhancement relative to Erikson et al. is the inclusion of structural variants. However, there is no easy way to evaluate the structural variant findings. Are they more or less than expected by chance or in-line with what is expected in the general population. The issue here is there is no matched control population with raw data processed in the same way as the MGRB cohort.

As noted by the Reviewer, there are challenges in obtaining the required comparison data to definitively answer this question. Recent efforts, such as gnomAD-SV, are an important step towards robust comparison in SV rates between the well elderly and a less selected population. However, currently the methodological complexities and lack of appropriate SV data have prevented such comparisons from being within the scope of the current study.

2.3 Similarly, the polygenic comparisons are concerning because of the difference in background population. While a reduction in polygenic risk is expected, its not clear what the influence of Australian vs UK vs a more heterogeneous gnomAD would be. It is known that there is population substructure that can lead to differences in polygenic risk, even among European individuals. An appropriate control would likely be Australian, not European.

We appreciate and agree with the Reviewer’s concerns on this point; the Editors also raised potential confounding from population background as an issue to address. Accordingly, we have invested significant effort in establishing that the polygenic score shifts we observed are highly unlikely to be due to differences in population structure or neutral (unrelated to the MGRB-depleted phenotypes) allele frequency divergence between the gnomAD, UKBB, and MGRB cohorts.

Before describing this new confirmatory analysis, we wish to highlight a key result in the original manuscript, in which we observed consistent depletion of risk alleles in MGRB relative to the other cohorts for depleted phenotypes only. This effect was extremely strong ($p < 1.1 \times 10^{-6}$ for both UKBB and gnomAD), indicating that the observed bias was very unlikely to be due to random allele frequency changes, and that instead the MGRB was consistently enriched for protective alleles, and depleted for risk alleles. Thus we believe it is strongly established that the MGRB is depleted for risk-associated alleles relative to UKBB and gnomAD, and only the related question of depletion of polygenic scores in MGRB remains to be addressed.

To do this we have followed a test procedure based on the Reviewer suggestion to use an Australian reference cohort. As noted, the ideal cohort to test for polygenic score shifts in MGRB due to depletion of cancer, cardiovascular disease, or neurodegeneration is a matched Australian reference cohort, of the same age and genetic background distribution, that has *not* been depleted for these clinical phenotypes. Unfortunately, such a reference cohort is not readily available and would be impractical to recruit at this stage. Given these issues, we elected to instead simulate the reference cohort.

In this simulation we consider the unmeasured Australian reference cohort (ARC) to be derived from the baseline cohort (eg UKBB), and the MGRB is in turn derived from the ARC (Figure 1a below). Differences in allele frequencies between ARC and the baseline cohort are assumed to be unrelated to the phenotypes under consideration; differences in allele frequencies between the MGRB and the ARC are due to selection against the MGRB-depleted phenotypes. Given that most loci are unrelated to these depleted phenotypes, the unmeasured distribution of allele frequency differences between the baseline cohort and ARC (d_1 in Figure 1a) across all loci is closely approximated by the measured distribution of allele frequency differences between the baseline cohort and the MGRB (d_2).

We can therefore bootstrap sample the ARC by starting with the baseline cohort allele frequencies, and perturbing them by an amount sampled from d_2 (Figure 1b). For each simulated ARC cohort a polygenic score (PS) can then be calculated, and compared to the observed PS in the MGRB. This procedure repeated over many bootstrap rounds forms a test for whether the observed difference in PS between MGRB and the baseline cohort (eg UKBB) can be explained by neutral allele frequency differences alone, or whether an additional factor (eg selection) is likely present.

a) Model

b) Test procedure

Figure 1: model (a) and test procedure (b) for the polygenic score comparisons. $d_2^{(k)}$ denotes bootstrap sample k of d_2 .

This test procedure is more comprehensive than that used in the original manuscript: the original test considered only sampling noise as a source of error, whereas this procedure

considers both sampling noise and neutral allele frequency differences. Accordingly we have replaced the original analysis in the manuscript with this updated procedure. The inclusion of an additional source of error has naturally increased the PS uncertainty, and thus the error bars in the revised manuscript Figure 1a are larger than those in the original submission.

Figure 1 above, with explanatory text, has been included in the manuscript as Supplementary Figure 11. Bootstrap distributions of the PS differences without rescaling have been provided as Supplementary Figure 4.

2.4 The somatic and mitochondrial analyses are quite interesting and novel. However, these processes, as stated by the authors, are heavily age-related. Without a age-matched control dataset, it is difficult to know whether the changes in any of these parameters are different in the well elderly vs normal elderly. The association with grip strength internal to the MGRB is interesting.

We agree this is an important area for follow-on work, and we are investigating methods by which we might measure some of our somatic biomarkers in the larger ASPREE cohort, without necessitating further large-scale whole-genome sequencing. However, given the early stage of development of these methods and the unavailability of an ideal Australian WGS control cohort, we believe this is a topic for future investigation.

Overall, the manuscript represents a useful dataset, with some concerns regarding the novelty and validity of findings.

Reviewer 3

This is a timely and interesting paper by Pinese et al. profiling the genetic characteristics of a cohort of “healthy elderly” who reached the age of 70 free of cancer, cardiovascular disease, and dementia. Timely because there are a number of papers now available investigating lifespan and healthspan, but these focus on common variation using genotyping arrays, and thus not only cannot analyse rare or structural variation in detail, but the approach taken by Pinese et al. also means that somatic variation can be analysed. This is the real novelty of this paper, in my opinion, and is deserved of publication. My comments and suggestions are included below. Note: I have not attempted to review the methods for the sequencing analysis, as this is beyond my area of expertise.

3.1. (Introduction, lines 81-84) Although the APOE locus has been consistently associated with lifespan and longevity, the FOXO3A locus is much less consistent. Both the Broer and Deelen 2014 GWAS (PMIDs 25199915 and 24688116) did not find strong evidence when comparing older individuals to younger controls (although the Broer paper claims it, the FOXO3A evidence is weak by comparison). Neither did Pilling or Timmers (2017, 2019) when looking at parents survival using UK Biobank and the LifeGen consortium, respectively (PMIDs 29227965 and 30642433). I suggest revising this section to reduce the emphasis on FOXO3A, and mention the loci arising from the Timmers paper especially (includes CDKN2A/B locus, HLA, etc.).

Thank you for bringing this to our attention; we have updated the manuscript.

3.2. (Results, lines 161-164) It is interesting that there are individuals in the cohort carrying potentially pathogenic variants. Is this prevalence comparable to or lower than other general studies (i.e. are they less common in healthy elderly)? I realise there is an analysis of pathogenic variants overall in the next section (starting line 188) but this does not seem to be the same question.

We have avoided direct cross-cohort comparisons for rare variants, due to the known technical challenges of avoiding cohort-specific false-positive and false-negative calls, related to sequencing artefacts and variant curation approaches. For this reason, we did not empirically compare the frequency of rare pathogenic variants in the MGRB directly to other cohorts. Previous studies have reported rates of pathogenic variation in unselected European cohorts of between 2.0% and 3.5%, which is slightly higher than our observed rate of 1.1%. However, given the differences in gene lists and variant interpretation, as well as the underlying technical difficulties, we believe that there is insufficient evidence to state that our observed rate is less than that reported by these prior studies. We have added a brief discussion of this point to the manuscript.

3.3. (Results, lines 188-207) Could this analysis be repeated using the genes associated with lifespan in the Timmer 2019 analysis (PMID 30642433)? I.e. they identified common variants associated with (parents) survival near ~12 genes, it would be interesting to know whether these genes carry fewer rarer/pathogenic variants in the healthy elderly than population controls. The authors could consider the same analysis for the loci identified by Zenin et al. in their GWAS of healthspan (PMID 30729179). Similar to the cancer/CAD loci already investigated, these loci have strong a priori evidence for involvement in healthy ageing.

Thank you for bringing these new studies to our attention; we have added them both to Figure 1.

3.4. Figure 1a. I think this would be improved by including the CAD results, even though they are null, for completeness (would be good to see estimate and CIs in the text at the very least). This would also help the general reader, to emphasise that it appears that rare variants for cancer are depleted, but no evidence was found (here) for CAD variants.

We agree that inclusion of a CAD arm to figure 1a would be valuable. Unfortunately, we do not have access to genotypes of a suitable platform-matched CAD-positive cohort (as were available for the cancer comparison, in the “45 and Up Study Cancer Cohort”), and so for the CAD comparisons were forced to rely on summary variant frequencies from the gnomAD database. The cancer and CAD statistics are therefore subtly different, and difficult to meaningfully compare or plot together.

More broadly, we are increasingly of the opinion that a rare variant burden analysis between technically mismatched cohorts (ie MGRB and gnomAD) is fraught with analytical difficulties. We included a caveat along these lines in the original text, by stating that the result was “potentially due to technical factors dominating differences in rare variant patterns between cohorts.” However, given the analytical questions and potential confusion engendered by this paragraph, we have decided to remove it from the manuscript, as we do not believe that it is materially important to our conclusions. We have also added a paragraph to the discussion that

emphasises the challenge of comparing rare variant burden between cohorts, and the need for further work in this area before any solid conclusions can be drawn.

We emphasise that the above concerns apply only to rare variants, which are enriched for platform-dependent false positive variants. The analyses that employ platform-matched cohorts (rare variant burden in cancer vs MGRB, somatic measures), or comparisons of common variation (Figure 1b-d) are less affected.

3.5. (Results, lines 227-230) Although it is stated in the methods, it might be useful to here recap what specific criteria were made for the MGRB. Is it here just referring to individuals with no cancer, cardiovascular disease, and dementia, or are the other criteria from the ASPREE study also referred to (which includes “no anaemia”, etc). To the casual reader it will be assumed to just be the former criteria, given the abstract.

We have now added this additional information to the manuscript, for clarity.

3.6. Statements about associations (significant or not) should include the relevant test statistic. For example Results section on polygenic scores. I realise some of this information is displayed graphically on Figure 1b but it should be available in the text, with additional information in the supplement. (Incidentally there is a typo on line 237, it should be Figure 1b not 2b). For example line 239 states the score for short lifespan (from the Deelen paper) “was significantly depleted in MGRB relative to gnomAD and UKB.” The authors should report by how much (and how significant) this difference was, otherwise it is impossible to determine whether this is meaningful. This may be better expressed the other way: “The MGRB healthy elderly harboured XX more lifespan-increasing alleles compared to gnomAD and UKB populations (beta, CIs, p...)” The figure contains information on a Normalised score (AU) which is useful to compare the different scores (i.e. to see that Atrial Fibrillation has a larger effect than breast cancer – if I am interpreting it correctly) but is not necessarily that interpretable.

We did not include measures of effect size (eg differences in mean polygenic score between MGRB, and UKBB or gnomAD) for these comparisons as we found them to be misleading: a strong depletion in disease phenotype translates to a very small (although still statistically significant) shift in **mean** disease risk. For example, consider the prostate cancer risk score of Hoffman *et al* 2015 (doi 10.1158/2159-8290.CD-15-0315). By Hoffman *et al*'s own estimates (Figure 4 in that paper), a risk score for prostate cancer based on 105 SNPs achieves a 6-fold difference in cancer odds between the highest and lowest score deciles in non-Hispanic white men -- a notable stratification of disease risk. However, when translated to the depletion setting, in which we compare the mean risk score between an unselected cohort and a cohort without prostate cancer, this manifests only an extremely subtle shift in mean score of around 0.026, or an apparent odds ratio of 1.026. A similar effect was exhibited in Lambert, Abraham, & Inouye 2019 (doi: 10.1093/hmg/ddz187, see Box 1), and Gibson 2012 (doi: 10.1038/nrg3118, see Figure 2a).

This disconnect between the predictive ability of a polygenic score, and the mean score shift that we used for our Figure 1, was our reason for excluding the mean score measures from the manuscript: we saw a risk that these measures would lead readers to conclude that the differences we observed were so subtle as to be meaningless, when in fact the scores were

predictive of disease state, particularly in the highest and lowest quantiles (see Figure 2 of our manuscript).

3.7. The section on somatic mutation accumulation with age is very interesting. Table 3, comparing the accumulation of different measures is the highlight of this analysis, accompanied with the analysis on functional outcomes (i.e. grip strength and gait). Given that the WGS is of blood, were any immune-relevant parameters available e.g. c-reactive protein, infections, lymphomas? The effect of mtDNA copy number on grip in men is very weak, so a more “relevant” outcome may be better powered. This section is again lacking detail however: lines 338 to 341 imply that all the somatic change measures were tested again grip and gait, yet these results are not presented. These should be included as a supplementary table.

Unfortunately we did not have access to any direct measures of immune function. We have recently observed a striking association between clonal haematopoiesis and a diagnosis of a hyperproliferative blood disorder (eg blood cancer, MDS), but believe that this is reflective of clonal outgrowth rather than age-related functional impairment. These data were only mature following submission and so were not included; it is a follow-on study that we are investigating using a larger and better-suited cohort. We have included all results from the somatic measure - functional outcome association testing in Supplementary Table 5.

3.8. Related to point 7. I think the “Summary” (lines 60-61) should better reflect the actual findings. Unless I have misinterpreted the text, it is misleading to state that “Pervasive age-related somatic changes were correlated with grip strength in men” when the p-values for mtDNA copy number~grip are very nominal (0.051 in 1st stage, 0.036 in 2nd) when clearly many statistical tests have been performed (5x somatic change measures, 2x outcomes, stratified by sex). These statements (and the accompanying conclusions/discussion) should be appropriately moderated.

We believe that the association between mtDNA copy number and grip strength, though subtle, is likely a real effect: although 18 tests were performed in the first stage, only a single comparison (mtDNA-grip strength in males) passed the first stage significance p value threshold of 0.1 and progressed to the second stage. Therefore the second stage p value of 0.036 should be reliable without correction.

This aside, we agree that the association is subtle in effect size, and accordingly we have removed “pervasive” from the Summary as we see it might convey an impression of strong effect. We reviewed our discussion of this result and feel it is appropriately moderated, with our discussion referring to a “potential measure” that “appeared to be associated”, followed by a call for further work. However, as we have mentioned to the Editor we are open to excluding the mtDNA link entirely if this section remains problematic.

Minor comments

3.9. It would be useful to include a title and some accompanying information at the start of each supplementary file, especially the tables. Currently it is unclear at points which table is which, and what the columns refer to.

We have added a “Description” tab to each supplementary table file, giving a brief description of the sheet contents, and a description of the content of each column.

3.10. "UK BioBank" just has one capitalized "B" i.e. should be "UK Biobank" (see www.ukbiobank.ac.uk)

Thank you, this has been corrected.

Reviewers' Comments:

Reviewer #1:

Remarks to the Author:

The authors have addressed most of my points sufficiently. However, there is one small point remaining.

In their rebuttal the authors mentioned that they added a PS based on the manuscript from Zenin et al. to Figure 1. However, this does not seem to be the case. Instead they have added a PS based on the manuscript from Pilling et al., but since this is based on a subset of the data (i.e. the UK Biobank part) from the manuscript from Timmers et al. it does not make much sense to have this in there. Hence, the authors should replace the PS based on the results from the manuscript from Pilling et al. with one that is based on the results from the manuscript from Zenin et al (and call it 'healthspan').

Reviewer #2:

Remarks to the Author:

The authors have adequately addressed my concerns.

Reviewer #3:

Remarks to the Author:

Thank you for the responses to my comments. My only reservation is with the analysis of mtDNA and grip (given the p-value in stage one was hardly convincing) however I am happy for the authors to include it in the manuscript, given the appropriate caution in interpretation now included.

REVIEWERS' COMMENTS

Reviewer #1 (Remarks to the Author):

The authors have addressed most of my points sufficiently. However, there is one small point remaining.

In their rebuttal the authors mentioned that they added a PS based on the manuscript from Zenin et al. to Figure 1. However, this does not seem to be the case. Instead they have added a PS based on the manuscript from Pilling et al., but since this is based on a subset of the data (i.e. the UK Biobank part) from the manuscript from Timmers et al. it does not make much sense to have this in there. Hence, the authors should replace the PS based on the results from the manuscript from Pilling et al. with one that is based on the results from the manuscript from Zenin et al (and call it 'healthspan'). We apologise for this mixup and have removed the Pilling et al PS result. We have not included the Zenin et al PS as only three of the Zenin SNPs pass our filtering criteria, and thus we expect the discriminatory capacity of the PS to be low. Despite the removal of the Pilling PS from this analysis we have retained our multiple testing penalisation for 26 tests (the original 13 PS x 2 comparator cohorts), as this is in truth the number of tests that were potentially reportable.

Reviewer #2 (Remarks to the Author):

The authors have adequately addressed my concerns.

Reviewer #3 (Remarks to the Author):

Thank you for the responses to my comments. My only reservation is with the analysis of mtDNA and grip (given the p-value in stage one was hardly convincing) however I am happy for the authors to include it in the manuscript, given the appropriate caution in interpretation now included.